

Climate
of the Past



# The triple oxygen isotope composition of phytoliths, a new proxy of atmospheric relative humidity: controls of soil water isotope composition, temperature, CO$_2$ concentration and relative humidity

**Clément Outrequin**[1], **Anne Alexandre**[1], **Christine Vallet-Coulomb**[1], **Clément Piel**[2], **Sébastien Devidal**[2], **Amaelle Landais**[3], **Martine Couapel**[1], **Jean-Charles Mazur**[1], **Christophe Peugeot**[4], **Monique Pierre**[3], **Frédéric Prié**[3], **Jacques Roy**[2], **Corinne Sonzogni**[1], **and Claudia Voigt**[1]

[1]CNRS, Aix Marseille Univ, IRD, INRA, CEREGE, Aix-en-Provence, France
[2]Ecotron Européen de Montpellier, Univ. Montpellier, UPS 3248 CNRS, Campus Baillarguet, Montferrier-sur-Lez, France
[3]Laboratoire des Sciences du Climat et de l'Environnement (LSCE/IPSL/CEA/CNRS/UVSQ), Gif-sur-Yvette, France
[4]Hydrosciences Montpellier, IRD, CNRS, Univ. Montpellier, Montpellier, France

**Correspondence:** Anne Alexandre (alexandre@cerege.fr)

**Abstract.** Continental atmospheric relative humidity is a major climate parameter whose variability is poorly understood by global climate models. Models' improvement relies on model–data comparisons for past periods. However, there are no truly quantitative indicators of relative humidity for the pre-instrumental period. Previous studies highlighted a quantitative relationship between the triple oxygen isotope composition of phytoliths, particularly the [17]O excess of phytoliths, and atmospheric relative humidity. Here, as part of a series of calibrations, we examine the respective controls of soil water isotope composition, temperature, CO$_2$ concentration and relative humidity on phytolith [17]O excess. For that purpose, the grass species *Festuca arundinacea* was grown in growth chambers where these parameters were varying. The setup was designed to control the evolution of the triple oxygen isotope composition of phytoliths and all the water compartments of the soil–plant–atmosphere continuum. Different analytical techniques (cavity ring-down spectroscopy and isotope ratio mass spectrometry) were used to analyze water and silica. An inter-laboratory comparison allowed to strengthen the isotope data matching. Water and phytolith isotope compositions were compared to previous datasets obtained from growth chamber and natural tropical sites. The results show that the $\delta'^{18}$O value of the source water governs the starting point from which the triple oxygen isotope composition of leaf water, phytolith-forming water and phytoliths evolves. However, since the [17]O excess varies little in the growth chamber and natural source waters, this has no impact on the strong relative humidity dependency of the [17]O excess of phytoliths, demonstrated for the 40 %–80 % relative humidity range. This relative humidity dependency is not impacted by changes in air temperature or CO$_2$ concentration either. A relative humidity proxy equation is proposed. Each per meg of change in phytolith [17]O excess reflects a change in atmospheric relative humidity of ca. 0.2 %. The $\pm15$ per meg reproducibility on the measurement of phytolith [17]O excess corresponds to a $\pm3.6$ % precision on the reconstructed relative humidity. The low sensitivity of phytolith [17]O excess to climate parameters other than relative humidity makes it particularly suitable for quantitative reconstructions of continental relative humidity changes in the past.

## 1 Introduction

The oxygen isotope composition of leaf water is an effective tool to trace processes at the soil–plant–atmosphere interface. It imprints the oxygen isotope composition of atmospheric CO$_2$ (Hofmann et al., 2017; Koren et al., 2019) and O$_2$ (Blunier et al., 2002; Brandon et al., 2020; Luz et al., 1999)

used to reconstruct changes in gross primary production. The leaf water also controls the oxygen isotope composition of organic and mineral compounds formed during the plant growth (Alexandre et al., 2012; Webb and Longstaffe, 2003, 2006). This is the case for phytoliths that are micrometric hydrous amorphous silica particles polymerized in abundance in plant tissues. Preserved in soils and sediments, phytoliths can be used for paleovegetation and paleoclimate reconstructions.

The main parameters influencing the isotope composition of bulk leaf water have been determined and modeled by numerous studies (Farquhar et al., 2007, and references therein; Cernusak et al., 2016). With regard to grasses, this composition can be estimated assuming a mixture between a pool of unevaporated water that circulate longitudinally in the veins and a pool of water that has been subject to evaporation in the intercellular space of the leaf stomata (Hirl et al., 2019; Leaney et al., 1985; Farquhar and Lloyd, 1993; Farquhar and Gan, 2003). The volume ratio of unevaporated vs. evaporated water in grass leaves, considered in the mixing equation, can be set at 0.2 (Alexandre et al., 2019; Hirl et al., 2019) and is independent of the transpiration rate. The $^{18}$O enrichment of the leaf water at evaporative sites can be estimated from the Craig and Gordon (1965) equation adapted to plant water (Eq. 4 from Cernusak et al., 2016, equivalent to Eq. 23 from Farquhar and Lloyd, 1993). The main variables of this equation are the isotope composition of the source water absorbed by the roots, equilibrium and kinetic fractionation factors between liquid water and water vapor, the isotope composition of atmospheric water vapor and the ratio of water vapor mole fractions in the atmosphere and at the leaf evaporation sites ($w_a/w_i$). This ratio is equal to the relative humidity (RH) when atmospheric and leaf temperatures (respectively $T_{atm}$ and $T_{leaf}$) are identical. Combining the Craig–Gordon equation with the mixing equation correctly depicts the trend of bulk leaf water $^{18}$O enrichment when RH decreases (Alexandre et al., 2018; Gan et al., 2003; Hirl et al., 2019). However, the estimated $^{18}$O enrichment values are often different from those observed by several ‰. This discrepancy indicates that our understanding of the parameters controlling changes in the $^{18}$O enrichment of leaf water is still incomplete.

The triple oxygen isotope composition of bulk leaf water, particularly its signature in $^{17}$O excess (as defined in Sect. 2), is influenced by fewer processes than $\delta^{18}$O alone (Aron et al., 2020; Luz and Barkan, 2010; Sharp et al., 2018). Indeed, the $^{17}$O excess of rainwater that feeds the soil water absorbed by the roots is less sensitive to temperature (Barkan and Luz, 2005; Uemura et al., 2010) or phase changes that occur during the air masses trajectories (Angert et al., 2003; Barkan and Luz, 2007; Landais et al., 2008; Uemura et al., 2010). As a result, the $^{17}$O excess of rainwater varies spatially much less than its $\delta^{18}$O (from $-30$ to 60 per meg when $\delta^{18}$O varies between $-25$‰ and $-5$‰; Aron et al., 2020). The variability range at a given location appears controlled by evaporative conditions at changing oceanic sources (Uechi and Uemura, 2019; Surma et al., 2021), continental vapor recycling and rain droplet evaporation (Landais et al., 2010; Li et al., 2015; Tian et al., 2019; Surma et al., 2021). The very few studies providing information on the variability of $^{17}$O excess in continental atmospheric vapor at low and middle latitudes (Lin et al., 2013; Surma et al., 2021; Ranjan et al., 2021) show that for a given location, it is in the same order of magnitude as that of rainwater, reflecting continental moisture recycling in addition to the evaporation conditions in the source region. Finally, for a given location, changes in the $^{17}$O excess of bulk leaf water should be primarily driven by local changes in the amplitude of the kinetic fractionation due to leaf water evaporation.

Recent calibration studies in growth chambers (Alexandre et al., 2018, 2019) and at a tropical site (Li et al., 2017) confirmed this assumption and highlighted a change of 2 to 4 per meg in the $^{17}$O excess of bulk leaf water for each percentage of RH. One of the growth chamber studies showed that this translates to a change of 3 to 5 per meg in phytolith $^{17}$O excess for each percentage of RH (Alexandre et al., 2018) and highlighted the potential of the $^{17}$O excess of fossil phytoliths to record past changes in continental atmospheric RH. Atmospheric RH is a major climate parameter. In combination with $T_{atm}$, it is used to estimate the concentration of the atmospheric water vapor, a major component of the water cycle and the main natural greenhouse gas (Held and Soden, 2000; Dessler and Davis, 2010; Chung et al., 2014). However, global climate models estimate RH from thermodynamic coupling between atmospheric water vapor and sea surface temperature (Bony et al., 2006; Stevens et al., 2017) and its actual variability is poorly accounted for. Model–data comparisons for the pre-instrumental period are necessary for models' improvement but face the lack of truly quantitative proxies of past RH. A promising proxy is the $\delta$D of plant biomarkers (Garcin et al., 2012; Sachse et al., 2012; Rach et al., 2017; Schwab et al., 2015; Tuthorn et al., 2015) recovered from buried soils and sediments. However, in addition to RH, the $\delta$D of plant biomarkers is dependent on other variables such as the $\delta$D in rainwater, the plant functional type and selective degradation of the biomarkers. The $^{17}$O excess of gypsum hydration that records the amplitude of surface water evaporation is also a promising new proxy of RH (Evans et al., 2018; Gázquez et al., 2018; Herwartz et al., 2017) but is limited to conditions favorable to gypsum formation. The $^{17}$O excess of phytoliths may hold the potential to complement the toolbox of proxies for RH reconstructions. However, it is necessary to assess whether and in which magnitude it can be impacted by environmental and climate parameters other than RH that changed over Quaternary glacial–interglacial cycles (Govin et al., 2014; IPCC, 2007). This applies to $T_{atm}$ that drives the difference between $T_{atm}$ and $T_{leaf}$ and hence the equilibrium fractionation between liquid water and water vapor, and the atmospheric concentration of $CO_2$ ($CO_2$) that drives the stomatal conductance and hence the kinetic fractionation in leaf water during

transpiration. The impact of changes in the triple oxygen isotope composition of soil water and atmospheric water vapor on the triple oxygen isotope compositions of leaf water and phytoliths also needs to be examined.

To address these issues, we present the results of a new growth chamber experiment in which the grass species *Festuca arundinacea* (*F. arundinacea*) was grown under variable conditions of RH, $T_{atm}$ and $CO_2$, mimicking past and present climate conditions in tropical areas. The setup was designed to monitor the evolution of the triple oxygen isotope composition of all the water compartments in the soil–plant–atmosphere continuum, particularly that of the atmospheric vapor. Phytolith isotope compositions were analyzed. The isotope data are compared to a previous growth chamber experiment and two datasets obtained from phytolith samples collected along a RH transect in western Africa (Alexandre et al., 2018) and a new isotope monitoring at the soil–plant interface carried out at the African Monsoon Multidisciplinary Analysis – Coupling the Tropical Atmosphere and the Hydrological Cycle (AMMA-CATCH) natural observatory in Benin (western Africa) (this study).

## 2 Fractionation and notation in the triple oxygen isotope system

In the triple oxygen isotope system, the mass-dependent fractionation factors between two phases, A and B ($^{17}\alpha_{A-B}$ and $^{18}\alpha_{A-B}$), are related by the exponent $\theta_{A-B}$ which is expressed as

$$^{17}\alpha_{A-B} = (^{18}\alpha_{A-B})^{\theta} \tag{1}$$

or

$$\theta_{A-B} = \ln^{17}\alpha_{A-B}/\ln^{17}\alpha_{A-B}. \tag{2}$$

The value of $\theta$ is equal to 0.529 for liquid water–water vapor equilibrium ($\theta_{equil}$; Barkan and Luz, 2005) and 0.518 for water vapor diffusion in air ($\theta_{diff}$; Barkan and Luz, 2007). While $\theta$ applies to a particular well-constrained physical process, the term $\lambda$ is used when several fractionation processes occur at the same time. In the $\delta'^{18}O$ vs. $\delta'^{17}O$ space, $\lambda$ represents the slope of the line linking $\delta'^{17}O_A - \delta'^{17}O_B$ to $\delta'^{18}O_A - \delta'^{18}O_B$ with

$$\lambda_{A-B} = (\delta'^{17}O_A - \delta'^{17}O_B)/(\delta'^{18}O_A - \delta'^{18}O_B) \tag{3}$$

and

$$\delta'^*O = \ln(\delta^*O + 1), \tag{4}$$

where $\delta^*O = (^*O/^{16}O)_{sample}/(^*O/^{16}O)_{standard} - 1)$. * stands for 18 or 17.

When water evaporates, three processes interplay in the leaf boundary layer, as conceptualized by the Craig and Gordon model (Craig and Gordon, 1965). These processes can

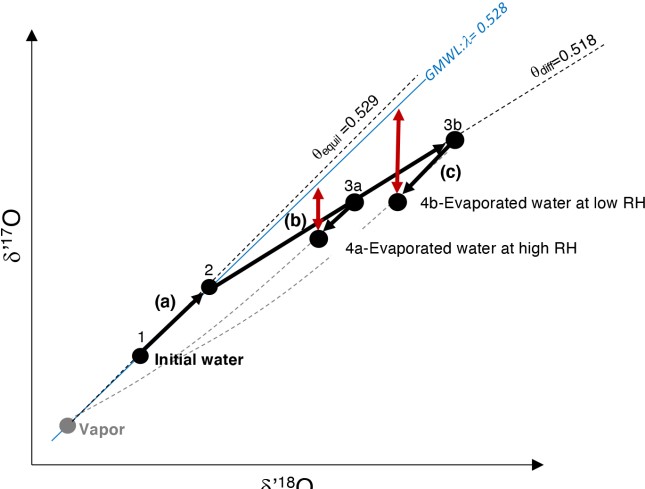

**Figure 1.** Representation of the three fractionating processes that interplay in the leaf boundary layer during evaporation, as conceptualized by the Craig and Gordon model: **(a)** from 1 to 2: equilibrium fractionation between initial water (unevaporated) and atmospheric vapor; **(b)** from 2 to 3a or 3b: fractionation due to vapor diffusion in humid (3a) or dry air (3b); **(c)** from 3a to 4a and 3b to 4b: fractionation due to exchange between evaporated water and atmospheric water vapor at high (3a to 4a) and low (3b to 4b) relative humidity. Red arrows: magnitude of the resulting $^{17}$O excess describing the departure of $d'^{17}O$ from a reference line with a slope $\lambda$ of 0.528, equivalent to the slope of the Global Meteoric Water Line (GMWL). $\theta_{equil}$: slope of the equilibrium line; $\theta_{diff}$: slope of the diffusion line.

be depicted in the $\delta'^{17}O$ vs. $\delta'^{18}O$ space as follows (Fig. 1): (a) water–vapor equilibrium fractionation, which is temperature dependent (Majoube, 1971), drives the isotope composition of leaf water along a line of a slope $\theta_{equil}$; (b) vapor diffusion in air, which is dependent on RH (or more specifically on $w_a/w_i$), drives the isotope composition of leaf water along a line of a slope $\theta_{diff}$; (c) exchange between leaf water and atmospheric water vapor, which decreases from turbulent to laminar and molecular vapor transport, and from high to low atmospheric water vapor concentration, drives the isotope composition of leaf water along a mixing curve linking the evaporated water and the atmospheric water vapor compositions. The combination of these processes leads to the $^{17}$O excess of evaporated leaf water, describing the departure of $\delta'^{17}O$ from a reference line with a slope $\lambda$ of 0.528, equivalent to the slope of the Global Meteoric Water Line (GMWL, expressed as $\delta'^{17}O = 0.528 \times \delta'^{18}O + 0.033$, Luz and Barkan, 2010). Thus, $^{17}$O excess is expressed as

$$^{17}O\text{-excess} = \delta'^{17}O - 0.528 \times \delta'^{18}O. \tag{5}$$

The difference of $^{17}$O excess between A and B is expressed hereafter by $^{17}$O excess$_{A-B}$. Since the value of 0.528 is close to the $\theta_{equil}$ value (0.529), measuring the $^{17}$O excess of water is very suitable for quantifying kinetic fractionation due to evaporation.

https://doi.org/10.5194/cp-17-1-2021

For the silica–water couple at equilibrium, the evolution of the exponent $\lambda$ ($\lambda_{\text{silica-water}}$) with temperature has been synthesized from data and can be calculated from Eq. (10) in Sharp et al. (2016). For the 5–35 °C temperature range, $\lambda_{\text{silica-water}}$ equals $0.524 \pm 0.0002$, in agreement with the theoretical $\theta_{\text{silica-water}}$ values (Cao and Liu, 2011).

## 3 Materials and methods

### 3.1 Growth chamber experimental setup

*F. arundinacea*, commonly referred to as tall fescue, is a globally distributed invasive grass species, widely used as forage (Gibson and Newman, 2001), and can adapt to a wide range of climatic conditions. Three growth chambers with a regulation of light intensity, $T_{\text{air}}$, RH and $CO_2$ were used to cultivate the tall fescue in three soil containers at the Microcosms platform of the Montpellier European Ecotron (France). The growth chambers as well as the growth protocol, described in detail in Appendix A, were adapted to allow for the monitoring of the isotope composition of all the water compartments in the soil–plant–atmosphere continuum (Fig. 2). The differences between the isotope composition of the water used for soil irrigation (6.55 ‰ and 29 per meg for $\delta'^{18}$O and $^{17}$O excess, respectively) and that of the water fogged into the chamber atmosphere (−5.64 ‰ and 17 per meg for $\delta'^{18}$O and $^{17}$O excess, respectively) were set close to the water liquid–vapor equilibrium fractionation value characteristic of natural systems (e.g., $\Delta'^{18}$O ranging from 9.65 ‰ to 9.06 ‰ between 20 and 28 °C; Majoube, 1971).

Three treatments which consisted in seven combinations of RH (40 %, 60 % and 80 %), $T_{\text{air}}$ (20, 24 and 28 °C) and $CO_2$ (200, 300 and 400 ppm) were applied to 18 fescue regrowth (two to three replicates per climate combination), as described in Table 1. At the end of each regrowth (lasting between 11 and 26 d), the fescue canopy was harvested. On $32 \pm 24$ g of fresh weight of fescue leaves (average per regrowth), 4 g were immediately inserted into glass gastight vials for leaf water extraction, while the remaining biomass was dried at 50 °C and kept for phytolith extraction. The humid air of each of the three growth chambers was continuously pumped via three heated lines and sent to a laser spectrometer for sequential analyses. The irrigation, soil and fogged waters were sampled before each regrowth for analyses. The volume of irrigation water consumed by transpiration was estimated (Table S1 in the Supplement).

### 3.2 Water vapor and liquid water triple oxygen isotope analyses

The humid air of the chambers was analyzed at Ecotron by wavelength-scanned cavity ring-down spectroscopy (CRDS) with a Picarro L2140-i spectrometer operated in $^{17}$O-excess mode.

For each chamber, the water vapor in the air was measured every second over a 420 min period before switching to the next chamber using a 16-port distribution manifold (Picarro A0311). After discarding the first 20 min to account for potential memory effects, the raw data were averaged over 80 min, resulting in five averages per vapor measurement period. Prior to each 420 min vapor measurement period, three working standards of liquid water were analyzed for calibration. This high calibration frequency allows to counteract a potential drift of the instrument. In order to estimate the background noise, the atmospheric water vapor fogged (without fractionation) from a constant water source into the three empty chambers was measured for each climate combination (except for the growth at 300 ppm $CO_2$) and two types of humidifiers. The precision on the means of the 80 min vapor measurements was 0.04 ‰ for $\delta^{18}$O and lower than 10 per meg for $^{17}$O excess (means of SD, $n = 19$).

The liquid water standard measurements necessary for the calibration of the water vapor measurements consisted of 10 injections per vial with the first six being discarded to account for memory effects. The dry air stream used for the liquid measurements was devoid of $CO_2$, contained less than 400 ppm of water vapor. The same dry air was used for flushing the growth chambers to limit measurement bias due to differences in the chemical composition of the analyzed growth chamber atmosphere and the dry gas used for calibration (Aemisegger et al., 2012; Brady and Hodell, 2021). The volumes of water standards vaporized to the spectrometer were adjusted to reach water vapor mixing ratios similar to those of the growth chamber atmospheres (i.e., between 12 000 and 30 000 ppm which corresponds to temperature / RH conditions of 24 °C / 40 % and 28 °C / 80 %). Thus, no correction for a mixing ratio dependency (e.g., Weng et al., 2020) was applied. The mean precision on the liquid water measurements for this mixing ratio range was 0.02 ‰ and 12 per meg for $\delta'^{18}$O and $^{17}$O excess, respectively (means of SD, $n = 21$). The variation for this range of mixing ratio was 0.04 ‰ and 7 per meg for $\delta'^{18}$O and $^{17}$O excess, respectively (SD of the means, $n = 21$).

The irrigation, soil and fogged water samples were analyzed at CEREGE (Centre de Recherche et d'Enseignement de Géosciences de l'Environnement, France) also using a Picarro L2140-i laser spectrometer (same configuration as at Ecotron). Each measurement run included three groups of three working standards for calibration, three replicates of an additional working standard for quality assurance (QA) and three replicates per water sample, with eight injections per vial. The two first injections were not considered. Each measurement was corrected for a memory effect (Vallet-Coulomb et al., 2021 for Rapid Communications in Mass Spectrometry). The precision on the QA value measurement was 0.02 ‰ for $\delta^{17}$O and $\delta^{18}$O and 11 per meg for $^{17}$O excess (1 SD; $n = 7$ measurement sessions). The averaged difference from the reference composition of the QA was 0.02 ‰,

**Table 1.** The growth chamber experiment: three RH, air temperature ($T_{air}$) and $CO_2$ concentration ($CO_2$) treatments (seven climate combinations and two to three replicates per combination), *F. arundinacea* regrowth duration, harvest fresh weight, productivity, phytolith concentration and percentage of long-cell (LC) phytolith types relative to long- and short-cell phytolith types.

| Treatments | | Sample[1] | Chamber (no.) | Container (no.) | RH (%) | $T_{atm}$ (°C) | $T_{leaf}$ (°C) | $CO_2$ (ppm) | VPD (kPa) | Regrowth duration (d) | Harvest fresh weight (g) | Productivity Av. (g d$^{-1}$) | Productivity SD | Phytolith conc. Av. (% d.w.) | Phytolith conc. SD | LC[2] (%) |
|---|---|---|---|---|---|---|---|---|---|---|---|---|---|---|---|---|
| RH | 40 | P2-40-12.07.17 | 2 | 2 | 40.1 | 24.0 | 22.0 | 400 | 1.78 | 20 | 32.8 | 1.6 | | 3.9 | | |
| | | P3-40-12.07.17 | 3 | 3 | 40.2 | 24.0 | 22.0 | 400 | 1.77 | 20 | 21.1 | 1.1 | | 3.2 | | |
| | | P4-40-12.07.17 | 4 | 4 | 40.0 | 24.0 | 22.0 | 400 | 1.78 | 20 | 34.4 | 1.7 | | 4.6 | | |
| | | | | | | | | | | | | 1.5 | 0.4 | 3.9 | 0.7 | 31 |
| | 60 | P2-60-12.06.17 | 2 | 2 | 59.5 | 24.0 | 22.0 | 400 | 1.20 | 14 | 15.1 | 1.1 | | 4.7 | | |
| | | P3-60-12.06.17 | 3 | 3 | 59.8 | 24.0 | 22.0 | 400 | 1.19 | 14 | 18.5 | 1.3 | | 2.6 | | |
| | | P4-60-12.06.17 | 4 | 4 | 59.5 | 24.0 | 22.0 | 400 | 1.20 | 14 | 14.2 | 1.0 | | 5.9 | | |
| | | | | | | | | | | | | 1.1 | 0.2 | 4.4 | 1.7 | 26 |
| | 80 | P2-80-07.08.17 | 2 | 2 | 79.1 | 24.0 | 22.0 | 400 | 0.62 | 26 | 108.4 | 4.2 | | 1.7 | | |
| | | P3-80-07.08.17 | 3 | 3 | 79.0 | 24.0 | 22.0 | 400 | 0.62 | 26 | 75.0 | 2.9 | | 1.1 | | |
| | | | | | | | | | | | | 3.5 | 0.9 | 1.4 | 0.4 | 16 |
| $T_{air}$ | 20 | P3-20-400-14.02.18 | 3 | 3 | 60.1 | 20.0 | 18.0 | 400 | 0.93 | 14 | 25.7 | 1.8 | | 2.4 | | |
| | | P4-20-400-14.02.18 | 4 | 4 | 60.1 | 20.0 | 18.0 | 399 | 0.93 | 14 | 38.4 | 2.7 | | 0.8 | | |
| | | | | | | | | | | | | 2.3 | 0.6 | 1.6 | 1.1 | |
| | 28 | P3-28-400-10.04.18 | 3 | 3 | 59.0 | 28.0 | 26.0 | 400 | 1.54 | 11 | 44.0 | 4.0 | | 3.2 | | |
| | | P4-28-400-10.04.18 | 4 | 4 | 59.3 | 28.0 | 26.0 | 400 | 1.53 | 11 | 20.2 | 1.8 | | 1.6 | | |
| | | P5-28-400-05.07.18 | 5 | 5 | 57.9 | 27.8 | 25.8 | 400 | 1.56 | 14 | 18.0 | 1.3 | | 2.6 | | |
| | | | | | | | | | | | | 2.4 | 1.4 | 2.4 | 0.8 | |
| $CO_2$ concentration | 200 | P3-24-200-16.03.18 | 3 | 3 | 60.1 | 24.0 | 22.0 | 200 | 1.18 | 14 | 15.7 | 1.1 | | 2.4 | | |
| | | P4-24-200-16.03.18 | 4 | 4 | 59.1 | 24.0 | 22.0 | 201 | 1.21 | 14 | 37.0 | 2.6 | | 1.9 | | |
| | | | | | | | | | | | | 1.9 | 1.1 | 2.2 | 0.4 | |
| | 300 | P3-24-300-30.03.18 | 3 | 3 | 60.0 | 24.0 | 22.0 | 300 | 1.19 | 14 | 17.7 | 1.3 | | 3.2 | | |
| | | P4-24-300-30.03.18 | 4 | 4 | 58.9 | 24.0 | 22.0 | 300 | 1.22 | 14 | 31.5 | 2.3 | | 2.5 | | |
| | | P5-24-300-10.08.18 | 5 | 5 | 58.5 | 24.0 | 22.0 | 300 | 1.23 | 15 | 20.5 | 1.4 | | 1.9 | | |
| | | | | | | | | | | | | 1.6 | 0.5 | 2.5 | 0.6 | |

[1] Sample name includes the sampling date (with DD.MM.YY). [2] Long-cell phytoliths/long- and short-cell phytoliths. Av.: average; SD: standard deviation.

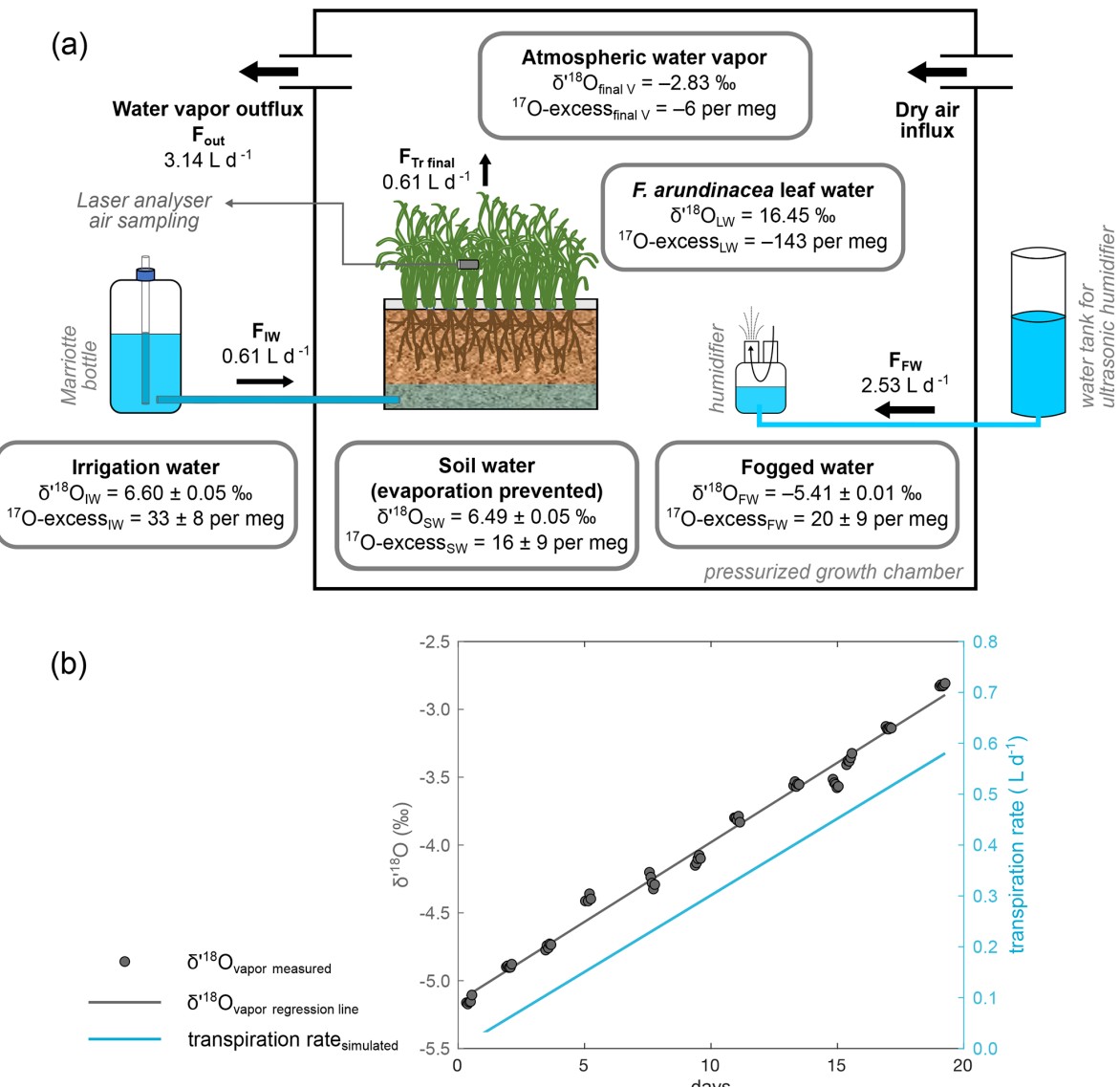

**Figure 2.** Scheme of the growth chamber setup for the isotope monitoring of the water compartments in the soil–plant–atmosphere continuum. **(a)** Isotope data are given for the final state of the P2-40-12.07.17 regrowth as an example (data from Table S1). The fogged water influx ($F_{fogW}$) is adjusted to keep a constant relative humidity (RH). $F_{fogW}$ final value is equal to the difference between the vapor outflux ($F_{out}$) and the irrigation water influx ($F_{IW}$), equivalent to the transpired water flux ($F_{Tr}$). $F_{fogW}$ is adjusted to keep a constant RH. **(b)** Linear correlation with the number of growing days of the atmospheric vapor $\delta'^{18}$O ($\delta^{18}O_v$) in the growth chamber. $\delta'^{18}$O values of the initial and final water vapor ($\delta'^{18}O_{initial\ V}$ and $\delta'^{18}O_{final\ V}$ in Table S1) were extrapolated from this correlation. The transpiration rate can be calculated on a daily basis using $\delta'^{18}O_v$ and an isotope mass balance as detailed in Table S1. The differences in $^{17}$O excess between irrigation water and soil water and between fogged water and atmospheric water vapor are due to methodological variability. When taking into account the totality of the climate combinations (Table S1), $\delta'^{18}$O and $^{17}$O-excess averages obtained for soil water ($6.28 \pm 0.16‰$ and $15 \pm 10$ per meg, respectively) and irrigation water ($6.50 \pm 0.06‰$ and $24 \pm 90$ per meg, respectively) or $^{17}$O-excess averages obtained for fogged water ($17 \pm 6$ per meg) and final atmospheric water vapor ($9 \pm 8$ per meg) are not significantly different (Student's $t$ tests).

$0.03‰$ and 1 per meg, for $\delta^{17}$O, $\delta^{18}$O and $^{17}$O excess, respectively.

Bulk water was extracted from the fescue leaves using a fluorination line and analyzed with dual-inlet isotope-ratio mass spectrometry (IRMS) (ThermoQuest Finnigan MAT 253 mass spectrometer) at LSCE (France), following the procedure previously detailed in Landais et al. (2006). The measurement precisions (2 runs of 24 dual-inlet measurements) were $0.1‰$ for $\delta^{18}$O and $\delta^{17}$O and 6 per meg for $^{17}$O excess.

All water isotope compositions were normalized on the Vienna Standard Mean Ocean Water – Standard Light Antarctic Precipitation (VSMOW-SLAP) scale, with an assigned

SLAP $^{17}$O-excess value of zero, following the recommendations of Schoenemann et al. (2013). A comparison between CEREGE, Ecotron and LSCE water measurements of the same laboratory standards was performed. Results are presented in Table B1. Differences between the laboratories were lower than 0.2‰, 0.1‰ and 10 per meg for $\delta^{18}$O, $\delta^{17}$O and $^{17}$O excess, respectively.

### 3.3 Phytolith extraction, counting and triple oxygen isotope analyses

Phytoliths were extracted from plants following a protocol detailed in Corbineau et al. (2013). Chemical removal of carbonate and organic compounds at 70 °C resulted in high-purity phytolith concentrates. Phytolith samples of 1.6 mg were then dehydrated under a flow of N$_2$ at 1100 °C (Chapligin et al., 2010) to prevent the formation of siloxane from silanol groups during dehydroxylation. Molecular O$_2$ was extracted using the IR laser-heating fluorination technique at CEREGE (Alexandre et al., 2006; Crespin et al., 2008; Suavet et al., 2010). At the end of the procedure, the gas was passed through a $-114$ °C slush to refreeze any molecule interfering with the mass 33 (e.g., NF potentially remaining in the line). Analyses of quartz and phytolith working standards with and without the slush showed that this step is essential to get reproducible $^{17}$O-excess data. The gas was directly sent to the dual-inlet mass spectrometer (ThermoQuest Finnigan Delta V Plus). Each gas sample was run twice. Each run consisted of eight dual-inlet cycles (integration time of 26 s). A third run was performed when the standard deviation (SD) on the first two averages was higher than 15 per meg for $^{17}$O excess.

The composition of the reference gas was determined against NBS28. The sample measurements were corrected on a daily basis using a quartz laboratory standard analyzed at the beginning of the day until a $^{17}$O-excess plateau was reached and again in the middle and at the end of the day. As recommended by Sharp and Wostbrock (2021), isotope compositions obtained from the analyses of NBS28 and six working standards (including phytolith standards) at CEREGE were compared to those obtained from the same standards measured at the University of New Mexico (UNM), where the data are directly calibrated relative to VSMOW-SLAP. The inter-laboratory comparison is presented in Table C1. The averaged CEREGE-UNM $^{17}$O-excess offset is $-25$ per meg and does not evolve with $\delta^{18}$O. This offset is comparable to the $\pm 24$ per meg average offset to VSMOW-SLAP standardized data obtained by other laboratories (Herwartz, 2021). Since this offset is larger than the data precision (15 per meg), the silica data obtained at CEREGE were corrected by adding 20 per meg to the measured $\delta^{17}$O value (which is equivalent to adding 20 per meg to the $\delta'^{17}$O or $^{17}$O-excess values). The data previously presented in Alexandre et al. (2018, 2019) were corrected accordingly. This correction

allowed robust comparisons between silica and water isotope compositions.

The reproducibility from $\delta^{18}$O and $^{17}$O-excess measurements of the quartz laboratory standard was 0.13‰ and 12 per meg, respectively (1 SD, $n = 21$ aliquots). For the phytolith samples, the averaged reproducibility from $\delta^{18}$O and $^{17}$O-excess measurements was 0.34‰ and 9 per meg, respectively (1 SD, 2 to 3 aliquots). The reproducibility from phytolith $^{17}$O-excess measurements was always lower than 15 per meg.

It was recently suggested that the dehydration step required before analyzing biogenic hydrous silica may induce an isotope fractionation biasing the $^{17}$O-excess values (Herwartz, 2021). Arguments for the absence of significant isotope fractionation during the high-temperature dehydration under N$_2$ flow step are presented in Appendix C.

Phytoliths obtained from three regrowths with different RH were mounted on microscope slides in Canada balsam for counting at a 600× magnification. The percentage of epidermal long-cell relative to short- and long-cell phytolith types was calculated (Alexandre et al., 2018). Two counts of the same assemblage gave a difference of 1 %.

### 3.4 Independent growth chamber and natural datasets used for comparison

To examine the effect of the soil water isotope composition on that of phytoliths, the results from the growth chamber experiment were compared with two datasets published in Alexandre et al. (2018): (1) the first dataset includes isotope compositions of leaf water and phytoliths of *F. arundinacea* grown at Ecotron in growth chambers where RH was varied in a similar way to the current RH treatments and $T_{atm}$ was set constant at 25 °C. In the setup of this growth chamber experiment, the isotope composition atmospheric water vapor was not measured. The isotope composition of the irrigation water ($-5.58$‰ and 26 per meg for $\delta'^{18}$O and $^{17}$O excess, respectively) was depleted relative to the recent growth chamber experiment but closer to natural waters. The fogged water had the same isotope composition as the irrigation water, similar to the one used in this study; (2) the second dataset includes isotope compositions of phytoliths extracted from soil tops collected along a vegetation and RH transect in western Africa. The vegetation is represented by savannas and humid forests and soil phytoliths are from grasses and trees. Rainfall $\delta'^{18}$O weighted monthly means at the sampling locations ranged from $-1.5$‰ to $-4.5$‰ (Online Isotopes in Precipitation Calculator – OIPC2-2; Bowen et al., 2005). Monthly mean RH extracted from the Climate Research Unit (CRU) 1961–1990 time series ranged from 57 % to 82 % (Alexandre et al., 2018).

Additionally, a new dataset was obtained from an ongoing monitoring at the AMMA-CATCH natural observatory (Galle et al., 2018) in Benin (western Africa) where dry forest and savannas prevail under a tropical humid climate. The

mean annual rainfall is 1300 mm with 60 % occurring during the rainy season between July and September. The mean annual $T_{atm}$ is 26 °C. RH ranges from 10 % to 20 % during the dry season and from 50 % to 80 % during the rainy season. The observatory is equipped with a meteorological station, providing hourly records of $T_{atm}$ and RH. The rainwater was collected continuously during the grass growing period and its isotope composition was analyzed (Sect. 3.2). Two grass species (*Hyparrhenia involucrata* and *Andropogon gayanus*) were collected at the beginning and in the middle of the rainy season in the dry forest and the savanna plots. Phytoliths were extracted from the stems that are supposed to be non-transpiring organs and analyzed (Sect. 3.3).

## 4  Results

### 4.1  Effectiveness of the growth chamber monitoring

For each growth, the water fluxes and the isotope compositions of the water compartments in the soil–plant–atmosphere continuum are presented in Table S1 and illustrated in Fig. 2a. The volumes of transpired water estimated by isotope mass balance are very close to the measured volumes (the slope of the correlation (not shown) is 1.0.; $r^2 = 0.97$), supporting the effectiveness of the monitoring. During the growth, the vapor $\delta'^{18}$O increases linearly (Fig. 2b), in response to the increasing contribution of transpired water to the atmosphere, unfractionated relative to the irrigation water (Welp et al., 2008). Since the $^{17}$O excess of the irrigation water is close to that of fogged water, the transpiration has little effect on the $^{17}$O excess of the atmospheric vapor. The mean value of $^{17}$O excess in atmospheric water vapor (Table S1) is statistically not different from that of the fogged water (Student's $t$ test).

### 4.2  Biomass productivity, transpiration and phytolith content

Considering all the regrowths, the biomass productivity and phytolith content range from $1.1 \pm 0.2$ to $3.5 \pm 0.9$ g d$^{-1}$ and from $1.4 \pm 0.4$ % to $4.4 \pm 1.7$ % dry weight, respectively (Table 1). The transpiration rate measured at the end of each regrowth ranges from 0.2 to 0.6 L d$^{-1}$ (Table S1). For each of these variables, there is a high variability between replicates which may explain the low correlation with the vapor pressure deficit (VPD). It is noteworthy that the averaged phytolith content is negatively correlated with the biomass productivity ($r^2 = 0.7$) and the transpiration rate ($r^2 = 0.4$). In the phytolith assemblages, the percentage of the long-cell phytolith type ranges from 16 to 31 %, increasing with VPD ($r^2 = 0.9$).

### 4.3  Leaf water isotope composition response to RH, $T_{air}$ and $CO_2$ changes

Evolutions of the triple oxygen isotope composition of leaf water with variable RH, $T_{air}$, $CO_2$ and the associated VPD (Table 2) are illustrated in Fig. 3. The 80 % to 40 % decrease of RH, corresponding to a 1.2 kPa increase of VPD, leads to a 5.9 ‰ increase in $\delta'^{18}$O and a 108 per meg decrease in $^{17}$O excess, which is characteristic of an evaporation trend. There is no impact of $CO_2$ changes on the $\delta'^{18}$O and $^{17}$O excess of leaf water. The increase in $T_{atm}$ from 20 to 28 °C, which corresponds to a 0.6 kPa increase of VPD, comes with a 5.1 ‰ decrease of $\delta'^{18}$O, which is opposite to an evaporative kinetic fractionation trend. The $^{17}$O excess does not change significantly with variable $T_{atm}$. The fractionation between irrigation and leaf water values is presented in Table 2. There is no significant trend in the triple oxygen isotope fractionation coefficient $\lambda$ between leaf water and irrigation water ($\lambda_{LW-IW}$) with RH.

### 4.4  Leaf phytolith isotope composition response to RH, $T_{air}$ and $CO_2$ changes

Changes in the triple oxygen isotope compositions of phytoliths with RH, $T_{air}$, $CO_2$ and the associated VPD values (Table S2) are illustrated in Fig. 3. Given the $\delta'^{18}$O and $^{17}$O-excess precision ranges, no changes are observed when $T_{atm}$ or $CO_2$ vary. The evaporation trend noted for leaf water is also observed for phytoliths but at a higher rate: the $\delta'^{18}$O of phytolith increases from $36.0 \pm 0.2$ ‰ to $48.6 \pm 1.4$ ‰ with RH decreasing from 80 % to 40 %. At the same time, the $^{17}$O excess of phytoliths decreases from $-177 \pm 2$ to $-336 \pm 22$ per meg.

The apparent fractionation between leaf water and phytoliths ($\delta'^{18}$O$_{Phyto} - \delta'^{18}$O$_{LW}$) is presented in Table 2. Between 80 and 60% RH, it is invariable ($\delta'^{18}$O$_{Phyto} - \delta'^{18}$O$_{LW}$ equals to $25.27 \pm 1.88$ ‰ and $26.77 \pm 0.35$ ‰ at 80 % and 60 % RH, respectively) but lower than expected for equilibrium fractionation between water and silica ($\delta'^{18}$O$_{silica} - \delta'^{18}$O$_w = 31$ ‰ and 33 ‰ for 26 and 18 °C, according to Dodd and Sharp, 2010). At 40% RH, $\delta'^{18}$O$_{Phyto} - \delta'^{18}$O$_{LW}$ is similar to the equilibrium value. The value of $\delta'^{18}$O$_{Phyto} - \delta'^{18}$O$_{LW}$ does not change significantly from 28 to 24 °C but decreases by 3.2 ‰ from 24 to 20 °C. This decrease is opposed to the increasing fractionation expected for equilibrium when temperature decreases. It is also opposed to the increase in the fractionation between irrigation and leaf waters ($\delta'^{18}$O$_{LW} - \delta'^{18}$O$_{IW}$). This translates into invariable fractionation between irrigation water and phytoliths ($\delta'^{18}$O$_{Phyto} - \delta'^{18}$O$_{IW}$). Overall, the $\lambda$ value calculated for the leaf water–phytolith couple ($\lambda_{Phyto-LW}$) averages $0.522 \pm 0.0005$ (Table 2).

### 4.5  AMMA-CATCH stem phytolith isotope composition

The isotope compositions of the stem phytoliths extracted from the two grass species collected at the AMMA-CATCH

**Table 2.** Triple oxygen isotope fractionation observed between bulk leaf water and irrigation water and phytolith and bulk leaf water of *F. arundinacea* for the three RH, $T_{air}$ and $CO_2$ concentration ($CO_2$) treatments.

| Treatment | | Sample[1] | RH (%) | $T_{atm}$ (°C) | $T_{leaf}$ (°C) | $CO_2$ (ppm) | VPD (kPa) | $\delta^{18}O_{LW}-\delta^{18}O_{IW}$ (‰) | Av. (‰) | SD (‰) | $\delta^{17}O_{LW}-\delta^{17}O_{IW}$ (‰) | $\gamma_{LW-IW}$ | Av. | SD | $\delta^{18}O_{Phyto}-\delta^{18}O_{LW}$ (‰) | Av. (‰) | SD (‰) | $\delta^{17}O_{Phyto}-\delta^{17}O_{LW}$ (‰) | Av. (‰) | SD (‰) | $^{17}O$ excess$_{Phyto-LW}$ (per meg) | Av. | SD | $\lambda_{Phyto-LW}$ |
|---|---|---|---|---|---|---|---|---|---|---|---|---|---|---|---|---|---|---|---|---|---|---|---|---|
| RH | 40 | P2-40-12.07.17 | 40.1 | 24.0 | 22.0 | 400 | 1.78 | 9.84 | | | 5.02 | 0.510 | | | 33.60 | | | 17.53 | | | −209 | | | 0.522 |
| | | P3-40-12.07.17 | 40.2 | 24.0 | 22.0 | 400 | 1.77 | 9.53 | 9.99 | 0.54 | 4.86 | 0.510 | 0.511 | 0.001 | 32.23 | 32.03 | 1.68 | 16.85 | 16.72 | 0.89 | −168 | −194 | 22 | 0.523 |
| | | P4-40-12.07.17 | 40.0 | 24.0 | 22.0 | 400 | 1.78 | 10.58 | | | 5.42 | 0.512 | | | 30.25 | | | 15.77 | | | −205 | | | 0.521 |
| | 60 | P2-60-12.06.17 | 59.5 | 24.0 | 22.0 | 400 | 1.20 | 6.60 | | | 3.35 | 0.508 | | | 26.38 | | | 13.77 | | | −163 | | | 0.522 |
| | | P3-60-12.06.17 | 59.8 | 24.0 | 22.0 | 400 | 1.19 | 7.35 | 7.04 | 0.39 | 3.75 | 0.510 | 0.510 | 0.002 | 27.04 | 26.77 | 0.35 | 14.12 | 13.97 | 0.19 | −152 | −165 | 13 | 0.522 |
| | | P4-60-12.06.17 | 59.5 | 24.0 | 22.0 | 400 | 1.20 | 7.18 | | | 3.68 | 0.512 | | | 26.90 | | | 14.02 | | | −179 | | | 0.521 |
| | 80 | P2-80-07.08.17 | 79.1 | 24.0 | 22.0 | 400 | 0.62 | 2.71 | 4.19 | 2.10 | 1.37 | 0.506 | 0.511 | 0.007 | 26.60 | 25.27 | 1.88 | 13.90 | 13.20 | 0.99 | −149 | −143 | 8 | 0.522 |
| | | P3-80-07.08.17 | 79.0 | 24.0 | 22.0 | 400 | 0.62 | 5.68 | | | 2.93 | 0.516 | | | 23.94 | | | 12.50 | | | −137 | | | 0.522 |
| $T_{air}$ | 20 | P3-20-400-14.02.18 | 60.1 | 20.0 | 18.0 | 400 | 0.93 | 11.49 | 11.05 | 0.62 | 5.97 | 0.520 | 0.520 | 0.001 | 22.59 | 23.59 | 1.42 | 11.74 | 12.27 | 0.75 | −188 | −184 | 5 | 0.520 |
| | | P4-20-400-14.02.18 | 60.1 | 20.0 | 18.0 | 399 | 0.93 | 10.61 | | | 5.51 | 0.519 | | | 24.59 | | | 12.80 | | | −181 | | | 0.521 |
| | 28 | P3-28-400-10.04.18 | 59.0 | 28.0 | 26.0 | 400 | 1.54 | 6.16 | | | 3.14 | 0.510 | | | 26.28 | | | 13.71 | | | −168 | | | 0.522 |
| | | P4-28-400-10.04.18 | 59.3 | 28.0 | 26.0 | 400 | 1.53 | 6.69 | 5.99 | 0.81 | 3.43 | 0.513 | 0.508 | 0.007 | 26.62 | 26.66 | 0.40 | 13.89 | 13.92 | 0.23 | −169 | −158 | 19 | 0.522 |
| | | P5-28-400-05.07.18 | 57.9 | 27.8 | 25.8 | 400 | 1.56 | 5.10 | | | 2.55 | 0.500 | | | 27.08 | | | 14.16 | | | −135 | | | 0.523 |
| $CO_2$ | 200 | P3-24-200-16.03.18 | 60.1 | 24.0 | 22.0 | 200 | 1.18 | 8.19 | 8.41 | 0.32 | 4.21 | 0.514 | 0.515 | 0.001 | 25.46 | 25.85 | 0.56 | 13.29 | 13.50 | 0.30 | −148 | −149 | 1 | 0.522 |
| | | P4-24-200-16.03.18 | 59.1 | 24.0 | 22.0 | 201 | 1.21 | 8.63 | | | 4.45 | 0.516 | | | 26.25 | | | 13.71 | | | −150 | | | 0.522 |
| | 300 | P3-24-300-30.03.18 | 60.0 | 24.0 | 22.0 | 300 | 1.19 | 6.46 | | | 3.30 | 0.511 | | | 26.47 | | | 13.82 | | | −157 | | | 0.522 |
| | | P4-24-300-30.03.18 | 58.9 | 24.0 | 22.0 | 300 | 1.22 | 10.03 | 8.10 | 1.80 | 5.18 | 0.517 | 0.513 | 0.003 | 24.13 | 25.65 | 1.32 | 12.60 | 13.39 | 0.69 | −143 | −150 | 7 | 0.522 |
| | | P5-24-300-10.08.18 | 58.5 | 24.0 | 22.0 | 300 | 1.23 | 7.80 | | | 3.99 | 0.513 | | | 26.35 | | | 13.76 | | | −151 | | | 0.522 |
| [2] 60% RH Av. | | | | | | | | 7.38 | | | | 0.511 | | | 26.23 | | | 13.70 | | | | | | 0.522 |
| [2] SD | | | | | | | | 1.10 | | | | 0.003 | | | 0.56 | | | 0.29 | | | | | | 0.0005 |

[1] Sample name includes the sampling date (with DD.MM.YY). [2] Average and standard deviation calculated for all experiments conducted at 60% RH. Av.: average; SD: standard deviation.

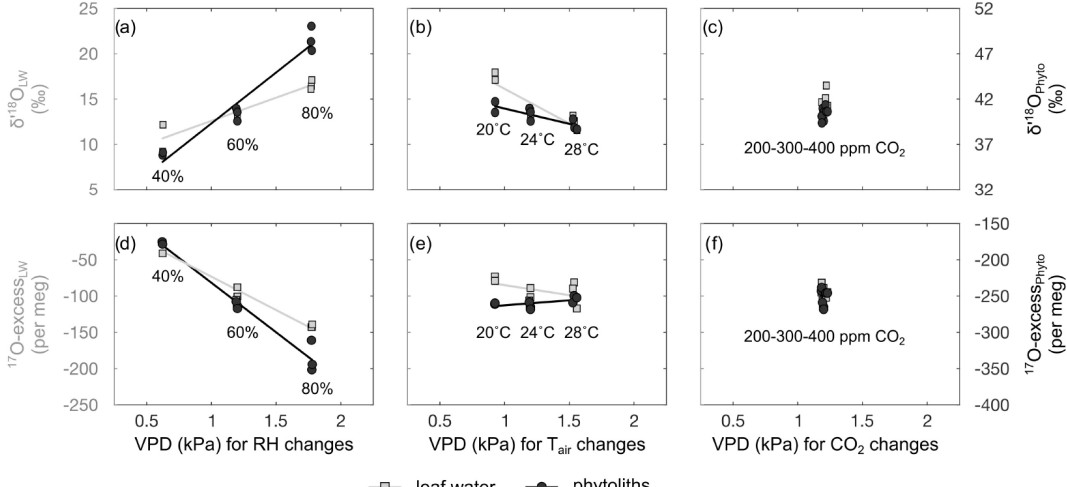

**Figure 3.** Impact of VPD changes associated with RH, $T_{air}$ and $CO_2$ concentration ($CO_2$) changes on $\delta'^{18}O$ and $^{17}O$-excess values of leaf water and phytoliths. Error bars on the measurement replicates are smaller than the symbols.

observatory are presented in Table 3. The isotope composition of the stem phytolith-forming waters is assumed to be close to the amount-weighed isotope composition of rainfall averaged for the growing season preceding the grass sampling. The obtained fractionation values for the rainwater–phytolith couple, i.e., $\delta'^{18}O_{Phyto} - \delta'^{18}O_{RW}$ and $\lambda_{Phyto-RW}$, are $31.18 \pm 0.63$ (1 SD) ‰ and $0.523 \pm 0.0004$, respectively. They are almost identical to the equilibrium values of $30.78 \pm 0.39$ ‰ and $0.524 \pm 0.0002$ calculated for $\delta'^{18}O_{silica} - \delta'^{18}O_{water}$ and $\lambda_{silica-water}$ after Dodd and Sharp (2010) and Sharp et al. (2016), respectively.

## 5  Discussion

### 5.1  Parameters controlling the triple oxygen isotope composition of leaf water

In order to consider the parameters responsible for the changes in the isotope composition of bulk leaf water, the $\delta'^{18}O$ and $^{17}O$-excess values were predicted for the current and 2018 growth chamber experiments using the modeling method described in Sect. 1. $T_{leaf}$ was assumed to be 2°C lower than $T_{atm}$ (Alexandre et al., 2019). For the 2018 experiment, since the vapor was not measured, the associated $\delta'^{18}O$ and $^{17}O$-excess values were set similar to those of the irrigation water as both fogged and irrigation waters came from the same tap water. The calculations are detailed in Table S3.

Figure 4 shows the observed and predicted $\delta'^{18}O$ and $^{17}O$-excess values of leaf water, for the $CO_2$, $T_{air}$ and RH treatments. For the $CO_2$ treatment, the small amplitude of changes in stomatal conductance, estimated from the $CO_2$ concentration (Barillot et al., 2010; Ainsworth and Rogers, 2007), has no effect on the predicted $\delta'^{18}O$ and $^{17}O$ excess of leaf water, in agreement with the observations. For the $T_{air}$

treatment, the model does not predict any significant change in $\delta'^{18}O$ and $^{17}O$ excess, in disagreement with the high $\delta'^{18}O$ value observed at 20°C. $T_{air-leaf}$ is difficult to measure or estimate, in growth chambers as well as outdoor, which certainly accounts for the inaccuracy frequently encountered when the $\delta'^{18}O$ of leaf water is modeled. However, in the present case, different-than-expected $T_{air-leaf}$ values would move the $\delta'^{18}O$ and $^{17}O$ excess of leaf water along the predicted line and cannot account for the offset in $\delta'^{18}O$ only observed at 20°C. For the current and 2018 RH treatments, the trends described by the $\delta'^{18}O$ and $^{17}O$ excess of leaf water when RH changes are correctly predicted, although some $\delta'^{18}O$ and $^{17}O$-excess values are significantly lower than expected from the model. For example, for the current RH treatment, the $\delta'^{18}O$ at 40 % is 6‰ lower than predicted. For the 2018 RH treatment, the $^{17}O$ excess can be 50 per meg lower than predicted. These model–data discrepancies could be related, as previously suggested (Alexandre et al., 2018; Li et al., 2017), to the misestimation of the isotope composition of the atmospheric water vapor. The isotope composition of the vapor was measured close to the leaves in the current growth chamber setup. However, it may be heterogeneous at the millimetric scale around the leaves. In addition, the high variability of the isotope composition of leaf water observed between replicates suggests that despite the precautions taken to harvest the grass regrowth as quickly as possible, changes in climatic parameters and isotope composition of the atmospheric water vapor when moving the containers outside the chambers may have a little biased the isotope compositions of the leaf waters. Outside atmospheric vapor with $\delta'^{18}O$ and $^{17}O$-excess values lower than inside the chamber would lower the $\delta'^{18}O$ and $^{17}O$ excess of the leaf water during the mixing step of the evaporation process (Sect. 2).

**Table 3.** $\delta'^{18}$O and $^{17}$O-excess values of stem phytoliths extracted from two grass species collected at the beginning and in the middle of the rainy season, at the AMMA-CATCH natural observatory in Benin (Djougou, stations of Naholou, 9.74281° N, 1.60635° E, and Bellefoungou, 9.78992° N, 1.71007° E). Averaged values of $\delta'^{18}$O and $^{17}$O excess for the rainwater of the rainy period preceding the samplings and values of fractionation between rainwater and phytoliths are shown. For comparison, values of fractionation between water and silica estimated according to Sharp et al. (2016) and Dodd and Sharp (2010) and assuming different values for $\lambda_{silica-water}$ are presented.

| Sample[1] | Vegetation | RH (%) | $T_{atm}$ (°C) | Phyto $\delta'^{18}$O Av. (‰) | Phyto $\delta'^{18}$O SD (‰) | Phyto $\delta'^{17}$O Av. (‰) | Phyto $\delta'^{17}$O SD (‰) | Phyto $^{17}$O excess Av. (per meg) | Phyto $^{17}$O excess SD (per meg) | n | RW $\delta'^{18}$O Av. (‰) | RW $\delta'^{17}$O Av. (‰) | RW $^{17}$O excess Av. (per meg) | Phyto–RW $\delta'^{18}O_{phyto}-\delta'^{18}O_{RW}$ (‰) | Phyto–RW $\delta'^{17}O_{phyto}-\delta'^{17}O_{RW}$ (‰) | Phyto–RW $^{17}$O excess$_{phyto-RW}$ (per meg) | $\lambda$ | $\delta'^{18}O_{silica}-\delta'^{18}O_{water}$ [3] (‰) | $\delta'^{18}O_{silica}-\delta'^{18}O_{water}$ [4] (per meg) | $\delta'^{18}O_{silica}-\delta'^{18}O_{water}$ [4] $\lambda=0.524$ (‰) | $^{17}$O-excess$_{silica-water}$ $\lambda=0.524$ | $^{17}$O-excess$_{silica-water}$ $\lambda=0.523$ | $^{17}$O-excess$_{silica-water}$ $\lambda=0.522$ | $^{17}$O-excess$_{silica-water}$ $\lambda=0.521$ (per meg) |
|---|---|---|---|---|---|---|---|---|---|---|---|---|---|---|---|---|---|---|---|---|---|---|---|---|
| NA-Hyp-T-17.09.18 | Savanna | 60 | 25 | 26.57 | 0.07 | 13.90 | 0.04 | −132 | 1 | 2 | −4.31 | −2.25 | 24 | 30.88 | 16.15 | −156 | 0.523 | 36.41 | −146 | 31.12 | −124 | −156 | −187 | −218 |
| BE-Andro-T-19.09.18 | Dry forest | 60 | 25 | 27.78 | 0.25 | 14.52 | 0.15 | −150 | 18 | 2 | −4.31 | −2.25 | 24 | 32.09 | 16.77 | −174 | 0.523 | 36.41 | −146 | 31.12 | −124 | −156 | −187 | −218 |
| NA-Hyp-T-19.05.19 | Savanna | 59.8 | 28.3 | 29.18 | 0.45 | 15.24 | 0.23 | −171 | 10 | 2 | −1.94 | −1.01 | 13 | 31.11 | 16.24 | −184 | 0.522 | 35.49 | −142 | 30.53 | −122 | −153 | −183 | −214 |
| BE-Andro-T-20.05.19 | Dry forest | 61.3 | 29.3 | 28.72 | 0.08 | 15.02 | 0.04 | −138 | 1 | 2 | −1.94 | −1.01 | 13 | 30.65 | 16.03 | −151 | 0.523 | 35.22 | −141 | 30.36 | −121 | −152 | −182 | −212 |
| Average | | | | | | | | | | | | | | 31.18 | | −166 | 0.523 | 35.88 | | 30.78 | | −154 | | |
| SD | | | | | | | | | | | | | | 0.63 | | 16 | 0.0004 | 0.62 | | 0.39 | | 2 | | |

[1] Sample name includes the grass species (Hyp for *Hyparrhenia involucrata* and Andro for *Andropogon goyanus*) and the sampling date (with DD.MM.YY). [2] Rainwater average amount weighted for the growth season preceding the sampling (1.5 and 5 months preceding September 2018 and May 2019, respectively). [3] According to Sharp et al. (2016). [4] According to Dodd and Sharp (2010). Av: average; SD: standard deviation.

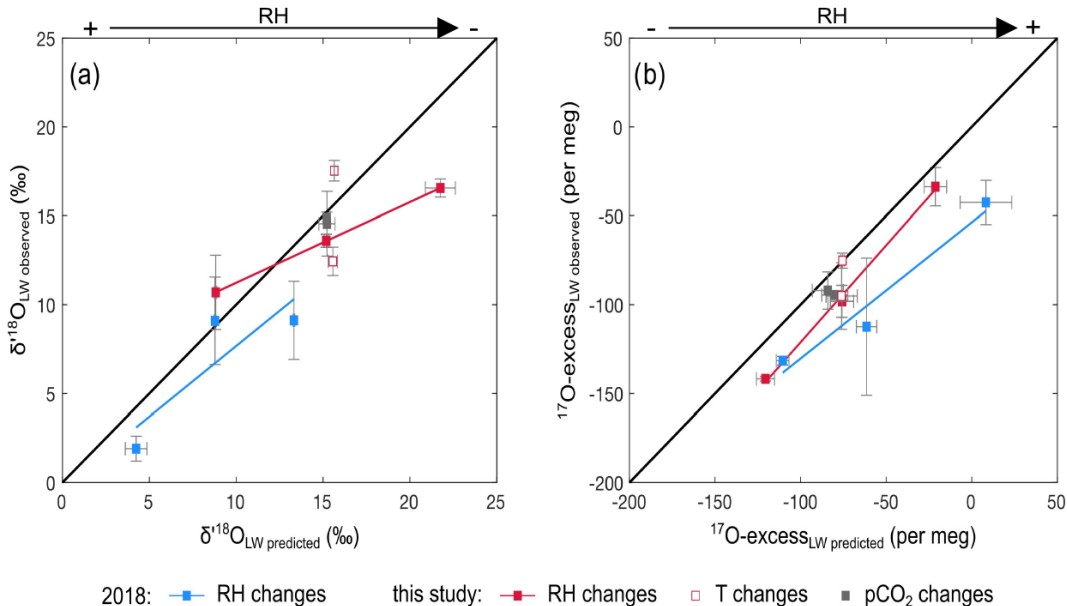

**Figure 4.** Observed vs. predicted $\delta'^{18}$O and $^{17}$O excess of leaf water for the RH, $T_{air}$ and CO$_2$ concentration (CO$_2$) treatments of the current experiment and the RH treatment of the 2018 growth chamber experiment (recalculated from Alexandre et al., 2018). Predicted $\delta'^{18}$O and $^{17}$O-excess values are reported in Table S3.

Figure 5 shows the leaf water isotope compositions at 40 %, 60 % and 80 % RH for the current and 2018 growth chamber experiments as well as the isotope compositions of the source water (i.e., irrigation water) in the $^{17}$O excess vs. $\delta'^{18}$O space. In agreement with the observations, sensitivity tests using the bulk leaf water model show that the isotope compositions of the source water (or the irrigation water) and the difference in composition between the source water and the atmospheric water vapor control the starting point from which the isotope composition of the leaf water evolves. When RH decreases, the isotope composition of the source water becomes the overriding factor. Because the $^{17}$O-excess values of the source waters in the current and 2018 experiments are similar, this has little effect on the dependency on RH of the $^{17}$O excess of leaf water.

This dependency is expressed for the current experiment by Eq. (6):

$$^{17}\text{O-excess}_{\text{LW}-\text{SW}} \; (\text{per meg}) = 2.8(\pm 0.2)\,\text{RH} \,(\%)$$
$$- 285(\pm 14) \; (r^2 = 0.97; \; p \text{ value} < 0.0001), \qquad (6)$$

where $^{17}$O-excess$_{\text{LW}-\text{SW}}$ is the difference of $^{17}$O excess between the LW and the source water (SW). Given the precision ranges, this equation is close to Eq. (7):

$$^{17}\text{O-excess}_{\text{LW}-\text{SW}} \; (\text{per meg}) = 2.1(\pm 0.7)\,\text{RH} \,(\%)$$
$$- 248(\pm 44) \; (r^2 = 0.55; \; p \text{ value} < 0.05), \qquad (7)$$

which is recalculated from the 2018 growth chamber dataset.

## 5.2 Parameters controlling the triple oxygen isotope composition of phytoliths

As observed in Fig. 3 and reported in Sect. 4, $T_{air}$ and CO$_2$ changes do not affect the $^{17}$O excess of phytoliths. With regard to $T_{air}$, this is the consequence of $\theta_{equil}$ and $\lambda_{silica-water}$ changing little with temperature. For instance, using an equilibrium $\delta'^{18}$O$_{silica}$ − $\delta'^{18}$O$_{water}$ value estimated from Dodd and Sharp (2010) and an equilibrium $\lambda_{silica-water}$ value of 0.524 (Sharp et al., 2016), the $^{17}$O excess of silica formed from a given water should vary by less than 10 per meg from 18 to 26 °C, which is lower than the analytical precision range.

The phytolith triple oxygen isotope compositions at 40 %, 60 % and 80 % RH are shown in Fig. 5 for the current and 2018 growth chamber experiments. The regression lines established from the isotope compositions are characteristic of evaporation trends. They are almost parallel for both experiments (Student's $t$ test). As for the leaf water, contrasting source water $\delta'^{18}$O values and differences between source water and water vapor $\delta'^{18}$O values can explain their offset in $\delta'^{18}$O. The isotope compositions of soil phytoliths from the 2018 natural transect sampling and stem phytoliths from the AMMA-CATCH site are also shown in Fig. 5. The isotope compositions of the soil phytoliths are distributed on and around the 2018 growth chamber line. The isotope compositions of the stem phytoliths are also located closer to the 2018 growth chamber line than to the current one. This can be explained by the proximity of the $\delta'^{18}$O values of rainwater at the natural sites (from −1.5 ‰ to −4.5 ‰ for the

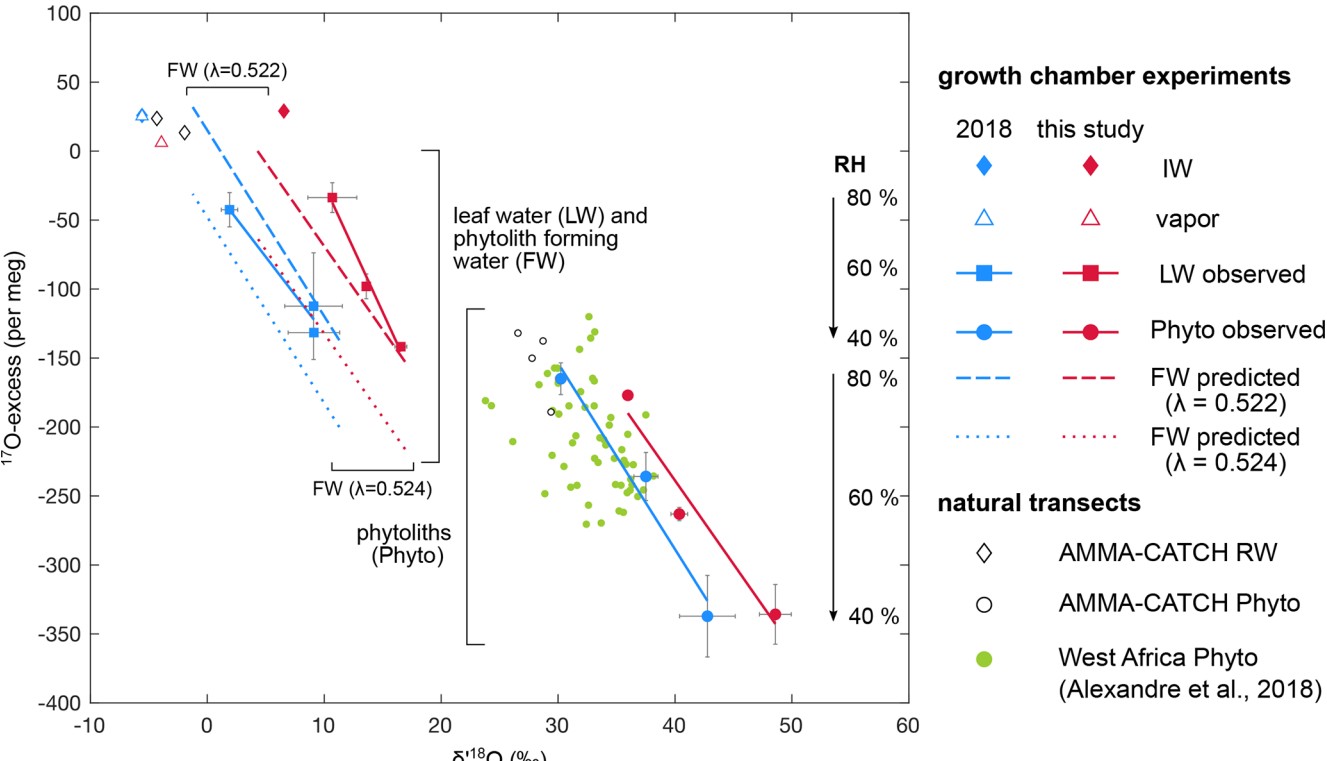

**Figure 5.** $^{17}$O excess vs. $\delta'^{18}$O of irrigation water (IW), final water vapor ($V$), bulk leaf water (LW), phytolith (Phyto) and phytolith-forming water (FW) observed and predicted for the current and 2018 RH treatment where RH varies from 40 % to 60 % and 80 %. Phytolith-forming water values are predicted using equilibrium $^{18}\alpha_{\text{silica−water}}$ estimated from Dodd and Sharp (2010) and $\lambda_{\text{silica−water}}$ values of 0.524 (Sharp et al., 2016) and 0.522 (Table S2). For comparison, values from the 2018 natural transect dataset (Alexandre et al., 2018) and from the AMMA-CATCH grass stem phytoliths and rainwater (RW) data (Table 3) are plotted.

natural transect (Alexandre et al., 2018) and from −1.9‰ to −4.3‰ for the AMMA-CATCH site) to the $\delta'^{18}$O of the irrigation water of the 2018 experiment (−5.6‰). These data support the idea that, as for bulk leaf water, the isotope composition of the source water and the difference in composition between source water and the water vapor govern the starting point from which the triple oxygen isotope compositions of grass phytoliths evolve with decreasing RH. Since the range of $^{17}$O-excess variation in the source waters and the atmospheric water vapor is narrow in the growth chambers and is expected to be narrow at natural sites, the source water should have little impact on the RH dependency of the $^{17}$O excess of phytoliths.

Figure 5 shows that the phytolith regression lines obtained from the 2018 and current growth chamber experiments are shifted in $^{17}$O excess, while this is not the case for the leaf water lines. This can be linked to the mean value of $\lambda_{\text{Phyto−LW}}$ calculated for the current growth chamber experiment ($0.522 \pm 0.0005$) that is slightly lower than the mean value recalculated for the 2018 experiment ($0.523 \pm 0.0012$; Table S2). Previous measurement and modeling of the evolution of the triple oxygen isotope composition of the grass leaf water and phytoliths with leaf length sug-

gested that $\lambda_{\text{Phyto−LW}}$ decreased from the base (0.523) to the apex (0.521) of the leaves (Alexandre et al., 2019). The mean value of the bulk grass leaf was 0.522 (recalculated from Alexandre et al., 2019). The increasing distance from the $\lambda_{\text{silica−water}}$ equilibrium value of 0.524 was interpreted as the sign of an apparent kinetic fractionation increasing from the base to the apex. The proximity of $\delta'^{18}O_{\text{Phyto}} − \delta'^{18}O_{\text{RW}}$ and $\lambda_{\text{Phyto−RW}}$ averaged values obtained for phytoliths from the grass stem samples collected at the AMMA-CATCH site (Sect. 4.4) to the $\delta'^{18}O_{\text{silica}} − \delta'^{18}O_{\text{water}}$ and $\lambda_{\text{silica−water}}$ equilibrium values rules out a fractionation occurring systematically during high-temperature dehydration. It rather suggests that the processes behind the apparent kinetic fractionation are limited when evaporation is restricted, as is the case in stems.

Heterogeneous silicification processes may contribute to an apparent kinetic fractionation between evaporated leaf water and silica. At least two patterns of silicification in grass leaf epidermis have been reported (Kumar et al., 2017). Short cells are among the first types of cells to be silicified, sometimes even before the leaf transpires. The process appears metabolically controlled and does not depend on the transpiration rate. To the contrary, long cells silicify only when

the plant starts to transpire and all the more so as the transpiration rate increases. If the isotope composition of leaf water changing with RH imprints only a portion of phytoliths, with the other portion formed from unevaporated water, then the apparent average values of $\delta'^{18}O_{Phyto} - \delta'^{18}O_{LW}$ and $\lambda_{Phyto-LW}$ will be lower than the values expected for isotope equilibrium. The magnitude of the resulting apparent fractionation will depend on the proportion of short-cell phytoliths and the difference of isotope compositions between evaporated and unevaporated leaf water. The two parameters vary with RH and VPD in opposite ways. Our data show that the distance between $\delta'^{18}O_{Phyto} - \delta'^{18}O_{LW}$ and equilibrium $\delta'^{18}O_{silica} - \delta'^{18}O_{water}$ values increases with RH, at the same time as the proportion of short-cell phytoliths (inverse of LC, Table 1 and Alexandre et al., 2018) increases, suggesting that, for the experimental conditions, the proportion of short-cell phytoliths has a preponderant influence.

The triple oxygen isotope compositions of the phytolith-forming water can be predicted at 40 %, 60 % and 80 % RH for the current and 2018 growth chamber experiments (Table S2), using $\delta'^{18}O_{Phyto} - \delta'^{18}O_{LW}$ and $\lambda_{Phyto-LW}$ values equivalent to the $\delta'^{18}O_{silica} - \delta'^{18}O_{water}$ and $\lambda_{silica-water}$ expected for equilibrium (Dodd and Sharp, 2010; Sharp et al., 2016). The regression lines from the predicted isotope compositions are shown in Fig. 5. For an equilibrium $\lambda_{Phyto-LW}$ value of 0.524 the phytolith-forming water lines are far from the leaf water ones. For a $\lambda_{Phyto-LW}$ value of 0.522, the phytolith-forming water lines are approaching the leaf water ones. At 40 % RH, the $^{17}$O excess of the phytolith-forming water is close to that of bulk leaf water. However, at 60 % and 80 % RH, the $^{17}$O excess of the phytolith-forming water is systematically higher than the $^{17}$O excess of bulk leaf water. At 80 % RH, it is even closer to the $^{17}$O excess of irrigation water than to that of bulk leaf water. The process described above may explain the offsets between phytolith-forming water and bulk leaf water $\delta'^{18}O$ and $^{17}$O-excess values, especially pronounced at high RH. It can also account for the low $\lambda_{Phyto-LW}$ values frequently encountered in phytolith studies. However, the imprecision of the measured $\lambda_{Phyto-LW}$ values that can vary by $\pm 0.001$ for a specific climate treatment (Table S2) makes it impossible to discern a trend of the apparent fractionation with RH or VPD.

### 5.3 The RH proxy

Whatever are the processes responsible for the low apparent $\lambda_{Phyto-LW}$ values, their effect on the RH dependency of the $^{17}$O excess of phytoliths appears weak or reproducible. This RH dependency can be expressed for the current growth chamber experiment by Eq. (8):

$$^{17}O\text{-excess}_{Phyto} \text{ (per meg)} = 4.1(\pm 0.3) \text{ RH (\%)}$$
$$- 500(\pm 18) \ (r^2 = 0.97; \ p \text{ value} < 0.0001), \tag{8}$$

that is indistinguishable from Eq. (9) expressed for the 2018 growth chamber experiment:

$$^{17}O\text{-excess}_{Phyto} \text{ (per meg)} = 4.4(\pm 0.5) \text{ RH (\%)}$$
$$- 510(\pm 33) \ (r^2 = 0.92; \ p \text{ value} < 0.001). \tag{9}$$

When applying Eq. (9) to calculate $^{17}$O excess$_{Phyto}$ from RH estimated for the 2018 natural transect (yearly averages of monthly RH means for months with at least 1 d of precipitation higher than 0.1 mm), the mean difference is $1 \pm 28$ per meg ($n = 55$). It is $12 \pm 28$ per meg when using Eq. (8).

Added to the fact that the $^{17}$O excess of phytoliths is insensitive to changes in the $\delta^{18}O$ of source water, $T_{air}$, $pCO_2$, or grass physiognomy (Alexandre et al., 2019), and assuming that variations in the $^{17}$O excess of soil water and atmospheric water vapor are narrow, this consistency between equations supports that in the 40 % to 80 % range, RH can be reconstructed from Eq. (10):

$$RH(\%) = 0.24 \pm 0.02 \ ^{17}O\text{-excess}_{Phyto} \text{ (per meg)}$$
$$+ 121 \pm 5 \ (r^2 = 0.98; \ p \text{ value} < 0.0001), \tag{10}$$

expressed for the current growth chamber experiment, and that is not different from Eq. (11):

$$RH(\%) = 0.21 \pm 0.02 \ ^{17}O\text{-excess}_{Phyto} \text{ (per meg)}$$
$$+ 112 \pm 6 \ (r^2 = 0.92; \ p \text{ value} < 0.0001), \tag{11}$$

expressed for the 2018 growth chamber experiment. For both equations, the statistical standard error associated with the RH estimates is close to 3 %. When applying Eq. (10) to calculate RH from $^{17}$O-excess$_{Phyto}$ values obtained for the 2018 natural transect, the mean difference between observations and estimates is $0.0 \pm 6.1$ % ($n = 55$). It is $2.7 \pm 6.6$ % per meg when using Eq. (11).

## 6 Conclusions

An inter-laboratory comparison (CEREGE-UNM) allowed us to standardize the phytolith triple oxygen isotope data relative to the VSMOW-SLAP scale. After this standardization, the apparent $\lambda_{Phyto-LW}$ values obtained so far vary between 0.521 and $0.523 \pm 0.0004$ (Alexandre et al., 2018, 2019). Heterogeneous silicification mechanisms in the leaves may result in an apparent kinetic fractionation and explain at least partly that $\lambda_{Phyto-LW}$ values are lower than the 0.524 value expected for equilibrium. When the forming water is well constrained as for phytoliths from non transpiring grass stems, the apparent $\lambda_{silica-water}$ value is $0.523 \pm 0.0004$, i.e., close to the equilibrium value.

The datasets presented for phytoliths, leaf waters and source waters show that the $\delta^{18}O$ value of the source water governs the position of the evaporation lines shown in the $\delta^{18}O$ vs. $^{17}$O-excess space for leaf water, phytoliths and phytolith-forming water. However, since the $^{17}$O excess of

the source waters varies little in growth chamber and in natural environment, the RH dependency of the $^{17}$O excess of phytoliths, verified for the 40 %–80 % RH range, is not affected. This RH dependency is not impacted either by the imposed $T_{air}$ and $pCO_2$ changes. The RH proxy equations (Eqs. 10 and 11) imply that each per meg of changes in $^{17}$O excess reflects a change in RH ranging from 0.19 % to 0.26 %. The $\pm15$ per meg reproducibility on the measurement of the $^{17}$O excess of phytoliths corresponds to a $\pm3.6$ % precision on the reconstructed RH. The low sensitivity of phytolith $^{17}$O excess to climate parameters other than RH makes it particularly suitable for quantitative reconstructions of continental RH changes in the past.

From there on, additional measurements of the triple oxygen isotope compositions of phytoliths, soil water and rainwater from sites located under different climate and covered by different vegetation types, as well as an examination of which RH (annual or seasonal, averaged over the growing period or over the growing period and the beginning of senescence) is captured by the $^{17}$O excess of phytolith assemblages extracted from sediments, will help to further specify the precision and the accuracy of this new RH proxy.

## Appendix A: Detailed description of the experimental protocol implemented in the growth chambers

Three growth chambers with a regulation of light intensity, $T_{air}$, RH and concentration of $CO_2$ ($CO_2$) were used to cultivate the grass *Festuca arundinacea* Schreb. in soil containers. These growth chambers had been adapted to allow for the monitoring of the isotope composition of the water compartments in the soil–plant–atmosphere continuum. In order to avoid isotope fractionation associated with water vapor condensation on the chamber walls, the chamber was flushed with $15\,m^3\,h^{-1}$ flow of dry air. This dry air was cooled down (or heated) to regulate the chamber $T_{air}$. Water vapor, produced without isotope fractionation from a reservoir of water of known isotope composition using an ultrasonic humidifier, was fogged into the chamber to reach the required RH. The humid air flow rate out of the chamber was maintained at a slightly lower value than the dry air inflow rate in order to create a slight overpressure in the chamber, preventing any outside air (and its water vapor) from entering the chamber. Pure $CO_2$ was injected into the chamber to reach and maintain the required $CO_2$.

Three plastic containers ($53 \times 35 \times 22\,cm\ L \times W \times D$) were filled with a 1.6 cm layer of quartz gravel on top of which 20 kg of dry commercial potting soil were added. A porous cup for water extraction was placed below the soil surface with its extraction tube hermetically extending outside of the container. The soil was humidified with a water of known isotope composition, also used for irrigation. The containers were sealed with a solid plastic lid to prevent soil evaporation. In total, 24 openings of $9 \times 0.8\,cm$ were cut into the lids and approximately fifty seeds of *F. arundinacea* were sown in each opening. During the germination period (from 18 to 21 d) the lids were covered with a transparent plastic film to keep a high RH around the seeds and to prevent evaporation. When most shoots reached a height of 4 cm, the plastic films were removed and silicone rubber (Bluesil™) was spread to fill up the openings, making sure it sealed the base of all the shoots. Elasticity of the rubber allowed the stems to grow in thickness while completely preventing soil evaporation. The containers were then introduced in the growth chambers to start the experiments. The *F. arundinacea* shoots were cut at 2 cm above the container lid, so that the regrowth took place under the imposed climate conditions. In each growth chamber, the soil was continuously irrigated from a Mariotte bottle (25 L) connected to the base of the container, so that a water saturation level of 5 cm was maintained at the base of the soil. Irrigation water in the Mariotte bottle was supplemented with $105\,mg\,L^{-1}$ of $SiO_2$ (in the form of $SiO_2\ K_2O$) and liquid paraffin was added on the water surface to prevent evaporation from the Mariotte bottle. The irrigation water was supplied from a large stock of $^{18}O$-enriched water. The $^{18}O$ enrichment was achieved by evaporation at $60\,°C$ under a flow of dry air and carried out so that the difference between the $\delta'^{18}O$ value of irrigation water and that of fogged water was

set close to the water liquid–vapor equilibrium fractionation characteristic of natural systems (e.g., $\delta'^{18}O_{liquid} - \delta'^{18}O_{vapor}$ ranging from 9.65‰ to 9.06‰ between 20 and 28 °C; Majoube, 1971).

After a regrowth of 11 to 26 d, the *F. arundinacea* canopy was cut at 2 cm above the lid. The harvested leaves averaged a fresh weight of $32 \pm 24\,g$. Next, 4 g were immediately inserted into glass gastight vials for leaf water extraction and the remnant/remaining biomass was dried at 50 °C and kept for phytolith extraction. Seven combinations of RH (40 %, 60 % and 80 %), $T_{air}$ (20, 24 and 28 °C) and $pCO_2$ (200, 300 and 400 ppm) were applied to the regrowths. The light was kept constant at $100\,\mu mol\,m^{-2}\,s^{-1}$ (provided by plasma lamps). For each combination, two to three replicates were run (Table 1). Due to constraints on the availability of the growth chambers, three to six regrowths were carried out successively on the same container. A failed growth, likely due to a lack of soil oxygenation, led to the preparation of a fourth container replacing one of the three initial containers. Since all the leaves of a regrowth were harvested at the end of each treatment, the interaction between the successive treatments on the same container should not affect the biomass productivity, the silicification pattern or the isotope composition of the bulk leaf water and phytoliths.

The air of each of the three growth chambers was continuously pumped via three $1/8''$ heated copper lines, at a flow rate of $0.5\,L\,min^{-1}$, and sent to a spectrometer for sequential analyses. The irrigation (Mariotte bottles), fogged (humidifier reservoir) and soil water were sampled before each regrowth and analyzed as described below. The volume of irrigation water consumed by transpiration was estimated by measuring the volume of water in the Mariotte bottle before and after each regrowth.

## Appendix B: Triple oxygen isotope analyses of water laboratory standards: inter-laboratory comparison

**Table B1.** Comparison between Ecotron, CEREGE and LSCE triple oxygen isotope measurements by CRDS and IRMS for three working standards of water.

| | CRDS-Ecotron Picarro L2140i | | | | | | | CRDS-CEREGE Picarro L2140i | | | | | | | IRMS-LSCE MAT 253 | | | Ecotron-LSCE | | | Ecotron-CEREGE | | | CEREGE-LSCE | | |
|---|---|---|---|---|---|---|---|---|---|---|---|---|---|---|---|---|---|---|---|---|---|---|---|---|---|---|
| | $\delta^{18}O$ | | $\delta'^{17}O$ | | $^{17}O$ excess | | $n$ | $\delta^{18}O$ | | $\delta'^{17}O$ | | $^{17}O$ excess | | $n$ | $\delta^{18}O$ | $\delta'^{17}O$ | $^{17}O$ excess | $\delta^{18}O$ | $\delta'^{17}O$ | $^{17}O$ excess | $\delta^{18}O$ | $\delta'^{17}O$ | $^{17}O$ excess | $\delta^{18}O$ | $\delta'^{17}O$ | $^{17}O$ excess |
| | ‰ Av. | ‰ SD | ‰ Av. | ‰ SD | per meg Av. | SD | | ‰ Av. | ‰ SD | ‰ Av. | ‰ SD | per meg Av. | SD | | ‰ | ‰ | per meg | ‰ | ‰ | per meg | ‰ | ‰ | per meg | ‰ | ‰ | per meg |
| GIENS-1 | −0.127 | 0.100 | −0.066 | 0.060 | 1 | 7 | 12 | −0.076 | 0.049 | −0.039 | 0.019 | 1 | 11 | 3 | −0.257 | −0.141 | −5 | 0.13 | 0.08 | 6 | −0.05 | −0.03 | 0 | 0.18 | 0.10 | 6 |
| ECO-1 | −5.676 | 0.100 | −2.972 | 0.050 | 29 | 4 | 12 | −5.578 | 0.003 | −2.924 | 0.015 | 25 | 13 | 3 | −5.610 | −2.938 | 28 | −0.07 | −0.03 | 1 | −0.10 | −0.05 | 4 | 0.03 | 0.01 | −3 |
| ICE-1 | −26.882 | 0.130 | −14.250 | 0.070 | 36 | 9 | 12 | −26.717 | 0.045 | −14.170 | 0.032 | 27 | 15 | 3 | −27.125 | −14.380 | 35 | 0.24 | 0.13 | 1 | −0.17 | −0.08 | 9 | 0.41 | 0.21 | −84 |

Av.: average; SD: standard deviation.

## Appendix C: Triple oxygen isotope analyses of silica

Isotope compositions obtained from the analyses of NBS28 and seven silica working standards at CEREGE were compared to those obtained from the same standards measured at the University of New Mexico (UNM), where the reference gas is calibrated relative to VSMOW-SLAP. The working standards are UWG-2 (garnet; University of Wisconsin, Valley et al., 1995), NM-SCO (olivine; UNM), NM-Q (quartz; Wostbrock et al., 2020), CEREGE-SCO (olivine; CEREGE), Boulangé 2008 (quartz; CEREGE), MSG50-900°C and P4-40-120717-900°C (phytolith standards dehydrated under $N_2$ flow; CEREGE). The values are presented in Table C1. When excluding SCO that is known to be heterogeneous in $\delta^{18}O$ (Suavet et al., 2009; Wostbrock et al., 2020), the averaged CEREGE-UNM $^{17}O$-excess offset is $-25$ per meg. The offset is independent from the $\delta^{18}O$ value of the analyzed sample. Since it is larger than the data precision (15 per meg), all the silica data obtained at CEREGE were corrected by adding 20 per meg to the measured $\delta^{17}O$ value (which is equivalent to adding 20 per meg to the $\delta'^{17}O$ or $^{17}O$-excess values).

It was recently suggested that when analyzing biogenic silica, the methods used for dehydration may bias the measured $^{17}O$-excess values (Herwartz, 2021). This should affect also the $\delta^{18}O$ value. It was indeed experimentally shown long ago that simple dehydroxylation at high temperature modifies the $\delta^{18}O$ value of hydrous silica. This was explained by incomplete removal of the exchangeable oxygens included in silanol groups, that get trapped in the silica structure when new siloxane groups form at high temperature (Labeyrie and Juillet, 1982). Based on this observation, several methods have been developed to fix the isotope composition of the exchangeable oxygen by isotope equilibration (Leclerc and Labeyrie, 1987) or remove all the exchangeable oxygen by stepwise fluorination (Chapligin et al., 2010; Haimson and Knauth, 1983; Leclerc and Labeyrie, 1987). An inter-laboratory comparison showed that there is a good agreement between the $\delta^{18}O$ values obtained from these different methods (Chapligin et al., 2011). The dehydroxylation under $N_2$ flow used today at CEREGE leads to $\delta^{18}O$ values for the biogenic working standards BFC (NERC), PS1772-8 (AWI) and MSG60 (CEREGE) of $28.3 \pm 0.6‰$ ($n = 7$), $43.0 \pm 0.3‰$ ($n = 6$) and $37.5 \pm 0.3‰$ ($n = 2$), respectively, in agreement with the inter-laboratory comparison pooled values ($29.0 \pm 0.3‰$, $42.8 \pm 0.8‰$ and $37.0 \pm 0.8‰$, respectively). The breaking of siloxane groups during the heating stage was also put forward as a process that could induce an isotope bias (Herwartz, 2021). In order to examine this assumption, the NBS28 quartz standard was analyzed at CEREGE after being heated at 900 °C under a $N_2$ flow. The aliquots processed in this way gave the same $\delta^{18}O$ and $^{17}O$-excess values ($9.727 \pm 0.227‰$ and $-51 \pm 12$ per meg ($n = 7$) for $\delta^{18}O$ and $^{17}O$ excess, respectively) as the unprocessed aliquots ($9.581 \pm 0.080‰$ and $-50 \pm 15$ per meg ($n = 4$), for $\delta^{18}O$ and $^{17}O$ excess, respectively). These data

argue for the absence of significant isotope fractionation during the biogenic silica dehydration step.

**Table C1.** Comparison between the triple oxygen isotope compositions of silica standards analyzed at CEREGE and the University of New Mexico.

| Standard | CEREGE uncorrected values | | | | | | | | | | UNM values calibrated relative to VSMOW and SLAP2 | | | | | | | | | CEREGE offset | | | CEREGE corrected values | | | | | |
|---|---|---|---|---|---|---|---|---|---|---|---|---|---|---|---|---|---|---|---|---|---|---|---|---|---|---|---|---|
| | ‰ vs. VSMOW | | | | | | per meg | | | | ‰ vs. VSMOW | | | | | | per meg | | | ‰ | per meg | | ‰ vs. VSMOW | | | | per meg | |
| | $\delta^{18}O$ | SD | $\delta^{17}O$ | SD | $\delta'^{18}O$ | $\delta'^{17}O$ | [17]O excess | SD | $\delta'^{17}O/\delta'^{18}O$ | n | $\delta^{18}O$ | SD | $\delta^{17}O$ | SD | $\delta'^{18}O$ | $\delta'^{17}O$ | [17]O excess | SD | $\delta'^{17}O/\delta'^{18}O$ | $\delta^{18}O$ | [17]O excess | $\delta'^{17}O/\delta'^{18}O$ | $\delta^{18}O$ | $\delta^{17}O$ | $\delta'^{18}O$ | $\delta'^{17}O$ | [17]O excess | $\delta'^{17}O/\delta'^{18}O$ |
| NM-SCO[a] | 6.874 | 0.200 | 3.550 | 0.109 | 6.851 | 3.543 | −74 | 3 | 0.517 | 2 | 5.320 | 0.160 | 2.750 | 0.080 | 5.306 | 2.746 | −55 | | 0.517 | 1.545 | −19 | | 6.874 | 3.570 | 6.851 | 3.563 | −54 | 0.520 |
| CEREGE SCO | 5.069 | 0.309 | 2.615 | 0.163 | 5.056 | 2.612 | −58 | 16 | 0.517 | 4 | | | | | | | | | | | | | 5.069 | 2.635 | 5.056 | 2.632 | −38 | 0.521 |
| UWG-2[a] | 5.716 | 0.129 | 2.950 | 0.070 | 5.700 | 2.945 | −64 | 13 | 0.517 | 5 | 5.700 | 0.057 | 2.940 | | 5.684 | 2.936 | −65 | 4 | 0.515 | 0.016 | 1 | 0.0013 | 5.716 | 2.970 | 5.700 | 2.965 | −44 | 0.520 |
| NBS28[a] | 9.581 | 0.080 | 4.977 | 0.055 | 9.535 | 4.965 | −70 | 15 | 0.521 | 4 | 9.570 | 0.055 | 4.991 | | 9.524 | 4.979 | −50 | 2 | 0.522 | 0.011 | −20 | −0.0012 | 9.581 | 4.997 | 9.535 | 4.984 | −50 | 0.523 |
| Boulangé 2008 50-100[b] | 16.252 | 0.208 | 8.440 | 0.108 | 16.121 | 8.405 | −107 | 13 | 0.521 | 100 | 15.825 | 0.059 | 8.240 | 0.033 | 15.701 | 8.206 | −84 | 2 | 0.523 | 0.421 | −23 | −0.0013 | 16.252 | 8.460 | 16.121 | 8.424 | −88 | 0.523 |
| NM-Q[a] | 18.045 | 0.219 | 9.374 | 0.103 | 17.884 | 9.330 | −113 | 13 | 0.522 | 3 | 18.070 | 0.136 | 9.419 | 0.072 | 17.909 | 9.375 | −81 | 5 | 0.523 | −0.024 | −32 | −0.0018 | 18.045 | 9.394 | 17.884 | 9.350 | −93 | 0.523 |
| MSG50-900°C[b] | 36.372 | 0.210 | 18.813 | 0.121 | 35.727 | 18.638 | −226 | 12 | 0.522 | 2 | 35.577 | | 18.447 | | 34.959 | 18.279 | −179 | 1 | 0.523 | 0.768 | −46 | −0.0012 | 36.372 | 18.833 | 35.727 | 18.658 | −206 | 0.522 |
| P4-40-120717 900°C[b] | 48.950 | 0.542 | 25.178 | 0.283 | 47.790 | 24.866 | −367 | 3 | 0.520 | 2 | 48.459 | 0.810 | 24.953 | 0.414 | 47.321 | 24.646 | −339 | 4 | 0.521 | 0.469 | −28 | −0.0005 | 48.950 | 25.198 | 47.790 | 24.886 | −347 | 0.521 |
| Offset average | | | | | | | | | | | | | | | | | | | | 0.277 | −25 | −0.0008 | | | | | | |
| SD | | | | | | | | | | | | | | | | | | | | 0.325 | 16 | 0.0011 | | | | | | |

[a] Sharp and Wostbrock (2021). [b] Measured at UNM for the present study. [c] Wostbrock et al. (2020). SD: standard deviation.

**Data availability.** Data may be extracted directly from the current article or requested from the corresponding author.

**Supplement.** The supplement related to this article is available online at: https://doi.org/10.5194/cp-17-1-2021-supplement.

**Author contributions.** CO and AA wrote the initial draft. AA, CO, JR, AL, CP, SD, CVC, CP and CV participated in the writing and editing of the final draft. CP, SD, JR, AA, CVC and AL conceptualized the growth chamber experiment and designed the methodology. CO, CP, SD and AA conducted the growth chamber experiment. AA, CO, CVC and CP designed and conducted the experiment at the AMMA-CATCH site. CP, AA, CO, AL, CVC, CS, MC, MP, FP and JCM performed the analyses. AA acquired the funding, and managed and supervised the project.

**Competing interests.** The authors declare that they have no conflict of interest.

**Disclaimer.** Publisher's note: Copernicus Publications remains neutral with regard to jurisdictional claims in published maps and institutional affiliations.

**Acknowledgements.** The authors express their gratitude to Zachary Sharp for the isotope measurements of the silica laboratory standards at the University of New Mexico. The authors also thank Theodore Ouani, Simon Afouda and Maxime Wubda (IRD-Benin) for their essential contribution at the AMMA-CATCH observatory. The authors thank the three anonymous reviewers.

**Financial support.** Our research was conducted in the frame of the HUMI-17 project supported by INSU-LEFE, the ANR (grant nos. ANR-17-CE01-0002-01), ECCOREV and labex OT-Med. It benefited from the CNRS human and technical resources allocated to the Ecotrons Research Infrastructure from the state allocation "Investissement d'Avenir" AnaEEFrance ANR-11-INBS-0001.

**Review statement.** This paper was edited by Alberto Reyes and reviewed by three anonymous referees.

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
