# Peer review of "The triple oxygen isotope composition of phytoliths, a new proxy of atmospheric relative humidity: controls of soil water isotope composition, temperature, CO2 concentration and relative humidity."

_Climate of the Past, 2021_

## Author Comment (AC1)

**Answer to Referee #1**

We thank the Referee #1 for his/her comments and respond below:

- **Lines 75 to 80. The authors should cite here recent studies that used triple oxygen and hydrogen isotopes in hydration water of minerals as a quantitative proxy for paleohumidity reconstructions, including Evans et al., 2018 and Gázquez et al., 2018. These studies are totally related to the final goals of this manuscript and should be cited as an example of quantitative RH proxy based on triple oxygen isotopes.**

In agreement with this comment the introduction of the manuscript will be modified as follows:

*Model-data comparisons for the pre-instrumental period are necessary for models' improvement but face the lack of truly quantitative proxies of past RH. A promising proxy is the $\delta D$ of plant biomarkers (Garcin et al., 2012; Sachse et al., 2012; Rach et al., 2017; Schwab et al., 2015; Tuthorn et al., 2015) recovered from buried soils and sediments. However, in addition to RH, the $\delta D$ of plant biomarkers is dependent on other variables such as the $\delta D$ in rainwater, the plant functional type and selective degradation of the biomarkers. The $^{17}O$-excess of gypsum hydration that records the amplitude of surface water evaporation is also a new promising proxy of RH (Evans et al., 2018; Gázquez et al., 2018; Herwartz et al., 2017) but is limited to conditions favorable to gypsum formation. The $^{17}O$-excess of phytoliths may hold the potential to complement the toolbox of proxies for RH reconstructions.*

- **In lines 108 to 114. I wonder if the author could translate this paragraph into a conceptual figure, explaining the sensitivity the isotope ratios to these parameters. Otherwise, it may be difficult to follow for non-specialized readers.**

In agreement with this comment, the following figure will be included in the manuscript:

[Figure]

*Figure 1: Representation of the three fractionating processes that interplay in the leaf boundary layer during evaporation, as conceptualized by the Craig and Gordon model: (a) from 1 to 2: equilibrium fractionation between initial water and atmospheric vapor; (b) from 2 to 3a or 3b: fractionation due to vapor diffusion in humid (3a) or dry air (3b); (c) from 3a to 4a and 3b to 4b: fractionation due to exchange between evaporated water and atmospheric water vapor at high (3a to 4a) and low (3b to 4b) relative humidity. Red arrows: magnitude of the resulting $^{17}O$-excess describing the departure of $\delta'^{17}O$ from a reference line with a slope $\lambda$ of 0.528, equivalent to the slope of the Global Meteoric Water Line (GMWL). $\theta_{equil}$: slope of the equilibrium line; $\theta_{diff}$: slope of the diffusion line.*

- **Sections 3.1 and 3.2. Did the air inlet to the chamber atmosphere come from the same cylinder as for the analyzer when doing the calibration with liquid waters? Did the instrument use Air Zero (dry synthetic air)?**

**Did you replace the air in the chambers with the same carrier? I am asking this because, in my personal experience, the use of different carrier gases (i.e. dry atmospheric air vs dry synthetic air) for calibration and for online measurements of water vapor can produce an offset in $^{17}O$excess. This needs to be clarified in this section.**

- **In section 3.2. Please, can you give the typical H2O concentrations measured with the CRDS analyzer from the chamber atmosphere? Did you consider/apply any linearity correction for the isotopic values? Did you take any measurement to monitor the drift of the instrument between calibrations?**

The atmospheric water vapor measurement protocol will be detailed to address these points in the method section as follows:

[revised manuscript text omitted]

---

## Author Comment (AC2)

**Answer to Referee #2**

We thank the Referee #2 for his/her comments and respond below:

- **I think a more nuanced discussion about the role and variability of vapor $^{17}$O-excess is warranted. There is very little published vapor $^{17}$O-excess data, so I don't think it is yet quite reasonable (as in line 62) to expect that vapor 17O-excess will vary little from place to place.**

In agreement with this comment, the sentence describing the $^{17}$O-excess data from water vapor will be specified as follows:

*The very few studies providing information on the variability of $^{17}$O-excess in continental atmospheric vapor at low and middle latitudes (Lin et al., 2013; Surma et al., 2021; Ranjan et al., 2021) show that for a given location, it is in the same order of magnitude as that of rainwater, reflecting continental moisture recycling in addition to the evaporation conditions in the source region.*

- **In thinking about this, it might be useful to draw comparisons between observations of vapor d-excess and expectations of vapor 17O-excess.**

Indeed, this would be very interesting to do for natural systems, with vapor measurements to back it up. However, this is beyond the scope of the present study. In the present study the isotope composition of the atmospheric water vapor is measured in the growth chambers to feed the Craig and Gordon model used to estimated the isotope composition of the leaf water. The mean values of $\delta'^{18}$O, $^{17}$O-excess and d-excess of the atmospheric water vapor in the growth chambers are now presented in sup. mat. 1.1. The mean values are fairly constant from an experiment to another (average and standard deviation of $-4.7 \pm 0.5$ ‰, $9 \pm 8$ per meg and $12.7 \pm 0.4$ ‰ for $\delta'^{18}$O, $^{17}$O-excess and d-excess) and not statistically different from the mean values of the fogged waters (Student's t-test). This clarification will be added in section 4.1:

*Since the $^{17}$O-excess of the irrigation water is close to that of fogged water the transpiration has little effect on the $^{17}$O-excess of the atmospheric vapor. The mean value of $^{17}$O-excess in atmospheric water vapor (Sup. mat.1.1) is statistically not different from that of the fogged water (Student's t-test)*

- **This study is limited to the tropics, but are there other regions with a limited range of vapor $^{17}$O-excess where a similar paleo-RH proxy might be worth exploring?**

As noted in section 5.1, *sensitivity tests using the bulk leaf water model show that the isotope compositions of the source water (or the irrigation water) and the difference in isotope composition between the source water and the atmospheric water vapor control the starting point from which the isotope composition of the leaf water evolves. When RH decreases, the isotope composition of the source water becomes the overriding factor. Because the $^{17}$O-excess values of the source waters in the current and 2018 experiments are close, this has little effect on the dependency on RH of the $^{17}$O-excess of leaf water.* In natural context, the difference in $^{17}$O-excess between rainwater and atmospheric water should be close to 10 per meg if equilibrium is reached. Source water evaporation and continental vapor recycling may additionally impact this difference, however additional field measurements are required to further assess the magnitude of the involved changes in $^{17}$O-excess of source water and atmospheric water vapor.

This will be nuanced in section 5. 2: *Since the range of $^{17}$O-excess variation in the source waters and the atmospheric water vapor is narrow in the growth chambers and is expected to be narrow at natural sites, both parameters should have little impact on the RH-dependency of the $^{17}$O-excess of phytoliths.* This will additionally be nuanced in section 5.3: *Added to the fact that the $^{17}$O-excess of phytoliths is insensitive to changes in the $\delta^{18}$O of source water, $T_{air}$, $pCO_2$, or grass physiognomy (Alexandre et al., 2019) and assuming that variations in the $^{17}$O-excess of soil water and atmospheric water vapor are narrow, this consistency between equations supports that in the 40 to 80% range, RH can be reconstructed from (eq. 10)*. Note that eq. 10 in the revised version is equivalent to eq. 11 in the version submitted to CPD.

- **Is the $\Delta'$ (or $\Delta'$, note the difference between the apostrophe and the prime notation and please be consistent throughout the manuscript) defined in Equation 3 necessary? This may be confusing for beginning readers because some recent triple oxygen isotope studies (e.g., Aron et al., 2021, Sharp et al., 2018, the 2021 RiMG book) have used the â ' notation rather than 17O-excess. If possible, I think it would be good to avoid the â symbol in this instance to minimize confusion.**

In agreement with this comment the $\Delta'^{17}O_{A-B}$ notation will be replaced in the manuscript (text and tables) by $\delta'^{17}O_A$ - $\delta'^{17}O_B$.

- **Section 3.2, paragraph 1: Did the authors account for potential memory effects in the vapor measurements when switching between ports on the manifolds? Were any measurements dropped or ignored just after switching to vapor measurements from a new chamber? Previous vapor isotope studies have shown that memory effects on vapor d18O and d-excess need to be accounted for when a this type of manifold setup is used (e.g., Simonin et al., 2013). I imagine that the sensitivity of 17O-excess to mixing makes this consideration important in this case as well.**

To address all the methodological points raised by the referees #1 and #2, the atmospheric water vapor measurement protocol will be detailed in the method section as follows :

*The humid air of the chambers was analyzed at Ecotron by Wavelength-Scanned Cavity Ring Down Spectroscopy (CRDS) with a Picarro L2140-i spectrometer operated in $^{17}O$-excess mode.*

*For each chamber, the water vapor in the air was measured every second over a 420 min period before switching to the next chamber using a 16-port distribution manifold (Picarro A0311). After discarding the first twenty minutes to account for potential memory effects, the raw data were averaged over 80 minutes, which resulted in 5 averages per vapor measurement period. Before each 420 min vapor measurement period, three working standards of liquid water were analyzed for calibration. This high calibration frequency allows to counteract a potential drift of the instrument. In order to estimate the background noise, the atmospheric water vapor fogged (without fractionation) from a constant water source into the three empty chambers was measured for each climate combination (except for the growth at 300 ppm $CO_2$) and two types of humidifiers. The precision on the 80 min vapor measurements was 0.04 ‰ for $\delta^{18}O_v$ and lower than 10 per meg for $^{17}O$-excess$_V$ (means of s.d., n=19).*

*The liquid water standards measurements necessary for the calibration of the water vapor measurements consisted of ten injections per vial with the first six being discarded to account for memory effects. The dry air stream used for the liquid measurements was devoid of $CO_2$, contained less than 400 ppm of water vapor and was the same as the one flushed in the growth chamber to reach the required RH (Appendix A). This should limit measurement bias due to differences in the chemical composition of the atmospheric water vapor analyzed and the dry gas used for calibration (Aemisegger et al., 2012). The volumes of water standards vaporized to the spectrometer were adjusted to reach water vapor mixing ratios similar to those of the growth chamber atmospheres (i.e. between 12000 and 30000 ppm which corresponds to temperature/RH conditions of 24°C/40% and 28°C/80%). The precision on the liquid water measurements for this range of mixing ratio was 0.02 ‰ and 12 per meg for $\delta^{18}O$ and $^{17}O$-excess, respectively (means of s.d., n = 21). The variation for this range of mixing ratio was 0.04 ‰ and 7 per meg for $\delta^{18}O$ and $^{17}O$-excess, respectively (s.d. of the means, n = 21). Thus, no correction for a mixing ratio dependency (e.g. Weng et al., 2020) was applied.*

- **Figure 1 provides a very useful schematic to understand the experimental setup. However, I have a few questions about the isotopic values reported. First, if evaporation is prevented from the soil, why are the soil d18O and soil 17O-excess values not identical (within analytical precision) to those of the irrigation water? Second, why is the 17O-excess of the final vapor (-6 per meg) so low? Is this a product of vapor mixing within the chamber? I encourage the authors to add d-excess data when possible (I assume this is accessible from the Picarro measurements) to explore the hydrologic processes that are going on in the chambers.**

The purpose of this study was to monitor the isotope composition of atmospheric water vapor in order to better constrain the Craig and Gordon model for estimating the isotope composition of leaf water. Therefore, the $^{17}O$-excess of the irrigation and fogged water were set close, which prevents to examine variations in $^{17}O$-excess in atmospheric water vapor in relation to transpiration and mixing processes.

For a given climate combination, there is no detectable trend regarding the $^{17}O$-excess in atmospheric water vapor. It is thus more appropriate to present the mean values of $\delta'^{18}O_{mean\ V}$ and $^{17}O$-excess $_{mean\ V}$ in figure 1 (presented below). $\delta'^{18}O_{mean\ V}$ and $^{17}O$-excess $_{mean\ V}$ values will be added in sup. mat. 1.1 for all the climate combinations.

The differences in mean isotopic composition between irrigation water and soil water and between fogged water and atmospheric water vapor are due to methodological variability. When taking into account the totality of the climate combinations (sup. mat. 1.1), $\delta'^{18}O$ and $^{17}O$-excess averages obtained for soil water (6.28 ± 0.16 ‰ and 15 ± 10 per meg, respectively) and irrigation water (6.50 ± 0.06 ‰ and 24 ± 9 per meg, respectively) are not significantly different

(Student's t-tests), confirming that no fractionation occurred during the vaporization. In the same way, $^{17}O$-excess averages obtained for fogged water (17 ± 6 per meg, respectively) and mean atmospheric water vapor (9 ± 8 per meg, respectively) are not significantly different (Student's t-tests). This will be added in caption of Figure 1.

The d-excess data were available for irrigation, soil water and atmospheric vapor, but not for leaf water (analyzed by IRMS), therefore, d-excess data for irrigation and soil waters are not presented since they don't add significant information. The average d-excess of the atmospheric vapor during the experiment remained very stable (d-excess = 12.5 ± 0.4 ‰) and similar to that of the fogged water (d-excess = 11.4 ± 0.5 ‰). The data will be presented in Sup.mat. 1.1.

[Figure]

***Revised Figure 1 (which will be figure 2 in the revised manuscript)**: Scheme of the growth chamber setup for the isotope monitoring of the water compartments in the soil-plant-atmosphere continuum. (a) Isotope data are given for the final state of the P2-40-120717 regrowth as an example (data from Sup. mat. 1.1). The vapor outflux in humid air ($F_{out}$) is equal to the sum of the fogged water infllux ($F_{fog\ W}$) and the irrigation water influx ($F_{IW}$) equivalent to the transpired water flux (T). $F_{fogW}$ is adjusted to keep a constant the relative humidity (RH). (b) Linear correlation with the number of growing days of the atmospheric vapor $d'^{18}O$ ($d^{18}O_v$) in the growth chamber. $d'^{18}O$ values of the initial and final water vapor ($d'^{18}O_{initial\ v}$ and $d'^{18}O_{final\ v}$ in Sup. mat. 1.1) were extrapolated from this correlation. The transpiration rate can be calculated on a daily basis using $d'^{18}O_v$ and an isotope mass balance as detailed in Sup. mat 1.1.*

*The differences in mean isotopic composition between irrigation water and soil water and between fogged water and atmospheric water vapor are due to methodological variability. When taking into account the totality of the climate combinations (sup. mat. 1.1), $d'^{18}O$ and $^{17}O$-excess averages obtained for soil water (6.28 ± 0.16 ‰ and 15 ± 10 per meg, respectively) and irrigation water (6.50 ± 0.06 ‰ and 24 ± 90 per meg, respectively) or $^{17}O$-excess averages obtained for fogged water (17 ± 6 per meg, respectively) and mean atmospheric water vapor (9 ± 8 per meg, respectively) are not significantly different (Student's t-tests).*

**References**

Aemisegger, F., Sturm, P., Graf, P., Sodemann, H., Pfahl, S., Knohl, A., and Wernli, H.: Measuring variations of δ 18O and δ 2H in atmospheric water vapour using two commercial laser-based spectrometers: An instrument characterisation study, Atmospheric Measurement Techniques, 5, 1491–1511, https://doi.org/10.5194/amt-5-1491-2012, 2012.

Alexandre, A., Webb, E., Landais, A., Piel, C., Devidal, S., Sonzogni, C., Couapel, M., Mazur, J.-C., Pierre, M., Prié, F., Vallet-Coulomb, C., Outrequin, C., and Roy, J.: Effects of leaf length and development stage on the triple oxygen isotope signature of grass leaf water and phytoliths: insights for a proxy of continental atmospheric humidity, 16, 4613–4625, https://doi.org/10.5194/bg-16-4613-2019, 2019.

Lin, Y., Clayton, R. N., Huang, L., Nakamura, N., and Lyons, J. R.: Oxygen isotope anomaly observed in water vapor from Alert, Canada and the implication for the stratosphere, PNAS, 110, 15608–15613, https://doi.org/10.1073/pnas.1313014110, 2013.

Surma, J., Assonov, S., and Staubwasser, M.: Triple Oxygen Isotope Systematics in the Hydrologic Cycle, Reviews in Mineralogy and Geochemistry, 86, 401–428, https://doi.org/10.2138/rmg.2021.86.12, 2021.

Weng, Y., Touzeau, A., and Sodemann, H.: Correcting the impact of the isotope composition on the mixing ratio dependency of water vapour isotope measurements with cavity ring-down spectrometers, 13, 3167–3190, https://doi.org/10.5194/amt-13-3167-2020, 2020.

---

## Author Comment (AC3)

**Answer to Referee #3**

We thank the Referee #3 for his/her comments and respond below:

**Authors Outrequin et al. submitted a manuscript about recent experiments investigating the controls over the triple oxygen isotope composition of phytoliths and the feasibility of using phytoliths as a paleo-aridity proxy. The authors detail a well thought out plant growth chamber experiment where temperature, carbon dioxide concentration, and humidity are each controlled. The authors conclude that relative humidity has the largest influence on the triple oxygen isotope value of the phytolith. The authors provide a new dataset from West Africa and examine the range in triple oxygen isotope values.**

**They compare their new results to previously published plant growth experiments and data from West Africa grasslands. It would be interesting to see values from different regions. However, the authors note in the conclusions that doing so is beyond the scope of the study. The only major critique of the paper is that the data from West Africa are not really described in terms of how it can be used to reconstruct relative humidity. The manuscript only notes that it follows closer to the 2018 growth experiment calculation due to the differences in the δ18O value of the initial water. It would be interesting to use Eq. 12 to predict the relative humidity in the modern analog (knowing the initial δ18O value of the precipitation water). Overall, this manuscript details a very time intensive and difficulty study and does a good job of distinguishing the main driver of the oxygen isotope composition of phytoliths. This manuscript is fitting for the journal and suitable for publication, pending addressing the major (optional) comment above and the small (and optional) comments below.**

The new dataset obtained from ongoing monitoring at the AMMA-CATCH Natural Observatory in Benin (West Africa) is limited to isotope composition data of stem phytoliths and rainwater. This data set is only used to examine the fractionation values for the rainwater-stem phytolith couples. This will be clarified in section 3.4 to avoid any misunderstanding. A more complete dataset (including the isotope compositions of soil water, leaf water and leaf phytoliths) is currently being processed and will be submitted for publication in the near future.

**Line 97: The denominator should be 18, not 17**
This will be corrected

**Figure 4: Are there any open red or blue circles? (Phyto predicted?) There are dotted lines but in the legend it says there are open red and open blue circles. May be worthwhile to add error bars on the phytolith measurements.**
The legend of this figure will be corrected and error bars added to the phytolith measurements (cf below).

[Figure]

***Revised Figure 5:*** *$^{17}$O-excess vs $\delta^{18}$O of irrigation water (IW), final water vapor (V), bulk leaf water (LW), phytolith (phyto) and phytolith-forming water (FW) observed and predicted for the current and 2018 relative humidity (RH) treatment where RH varies from 40 to 60 and 80%. Phytolith-forming water values are predicted using equilibrium $^{18}\alpha_{Silica\text{-}water}$ estimated from Dodd and Sharp (2010) and $\lambda_{Silica\text{-}water}$ values of 0.524 (Sharp et al., 2016) and 0.522 (Sup. mat. 1.3). For comparison, values from the 2018 natural transect dataset (Alexandre et al., 2018) and from the AMMA-CATCH grass stem phytoliths and rainwater (RW) data (Table 3) are plotted.*

**Why not add Eq. 12 and predict relative humidity of the natural phytolith samples?**

In agreement with this comment, the prediction will be added in section 5.3. When applying (eq. 11) to calculate RH from $^{17}$O-excess$_{Phyto}$ values obtained for the 2018 natural transect, the mean difference is 0.0 ± 6.1 % (n=55). It is 2.7 ± 6.6 % per meg (n=55) when using (eq. 12).